# SEMANTIC ANCHORING IN LLMS: THRESHOLDS, TRANSFER, AND GEOMETRIC CORRELATES

## ABSTRACT

We propose *semantic anchoring*, a unified account of how large language models turn pretrained capacity into goal-directed behavior: external structure (in-context examples, retrieval, or light tuning) binds the model's latent patterns to desired targets. Unified Contextual Control Theory (UCCT) formalizes this via anchoring strength $S = \rho_d - d_r - \log k$, where $\rho_d$ measures target cohesion in representation space, $d_r$ measures mismatch from prior knowledge, and $k$ is the anchor budget. UCCT predicts threshold-like performance flips and strictly generalizes in-context learning, reading retrieval and fine-tuning as anchoring variants. Three controlled studies provide evidence. Experiment 1 demonstrates cross-domain anchoring rebinding strong priors in text and vision. Experiment 2 varies representational familiarity via numeral bases (base-10/8/9) at fixed complexity, yielding ordered thresholds and transfer patterns tracking $\rho_d$, $d_r$, and $S$. Experiment 3 establishes a geometry-to-behavior correlate: layer-wise peak anchoring and trajectory area predict few-shot thresholds $\theta_{50}$. UCCT offers testable theory and practical metrics for optimizing prompts, retrieval, and tuning.

## 1 INTRODUCTION

Large language models (LLMs) can solve new tasks after seeing just a few examples, yet they also fail unpredictably on closely related prompts. Where does reliable, goal-directed behavior come from if it is not explicitly programmed?

A minimal prompt change can flip behavior. Ask `2 - 3 = ?` and the model returns $-1$. Prepend two examples,

```
Example 1: 2 - 3 = 5
Example 2: 7 - 4 = 11
Question: 15 - 8 = ?
```

and mainstream LLMs switch to 23, as if "−" were rebound to a new rule. Parameters did not change; the *binding* did. Such abrupt reinterpretations suggest pretraining stores *latent* patterns that coherent anchors can bind (or misbind) to targets.

**A working metaphor (baited anchors).** Treat the pretrained model as an ocean of latent patterns. An *anchor* is lowered, but what bites depends on the *bait*: with no bait, behavior follows prior currents; with incoherent bait, it misbinds; with coherent bait, the intended pattern hooks. The goal is *just enough* bait to flip behavior without attracting off-target structure.

**What we mean by in-context learning (ICL).** ICL is *test-time* adaptation from prompt or retrieved context (examples, instructions, tools) with *no* parameter updates (Brown et al., 2020); format alone can dominate label use (Min et al., 2022). Behavior depends on $p_\theta(y \mid x, \text{context})$, where $y$ is the model output, $x$ is the input query, and context steers intermediate representations rather than gradients. In our lens, ICL is *anchoring*: external structure biases allocation over latent pattern clusters, producing threshold-like changes when anchoring strength crosses a critical value.

**Our theory.** We propose the Unified Contextual Control Theory (UCCT), in which pretraining stores latent patterns $P_{\text{prior}}$ (the model's prior knowledge) that acquire task semantics when *semantic anchors* (in-context examples or instructions) bind them to target patterns $P_T$ (the desired task

behavior). UCCT adopts a two-stage Bayesian view and defines anchoring strength as

$$S \;=\; \rho_d \;-\; d_r \;-\; \log k,$$

where $\rho_d$ measures how tightly the target pattern $P_T$ clusters in representation space (cohesion), $d_r$ measures how far $P_T$ is from the prior $P_{\text{prior}}$ (mismatch), and $k$ is the number of anchor examples provided. When $S$ exceeds a task-dependent threshold $S_c$, performance shifts abruptly, consistent with threshold-like capability changes under scale or context (Wei et al., 2022; Schaeffer et al., 2023). We estimate $S$ with calibrated proxies (whitening and $z$-scaling on a dev set; details in Appx. D).

**What is new.** UCCT *strictly generalizes ICL* and reads retrieval-augmented generation and fine-tuning as the same anchoring process acting on one measurable score $S$: retrieval raises effective cohesion $\rho_d$, fine-tuning reduces mismatch $d_r$, and few-shot adjusts the example budget $k$. We provide controlled tests that fix computational complexity while varying *representational familiarity*, and we uncover a geometry-to-behavior *correlate* that summarizes anchoring trajectories.

**Why** UCCT**?** Existing work explains *that* ICL transitions occur (Wei et al., 2022; Park et al., 2024) and *where* geometry shifts (Park et al., 2025), but not *when* behavior flips for a specific prompt or *how much* anchor budget is needed. UCCT fills this gap with a measurable predictor: $S = \rho_d - d_r - \log k$ quantifies the three forces (cohesion, mismatch, budget) and predicts shot midpoints $k_{50}$ across different bases, tasks, and models. This enables practical optimization (e.g., "add 2 more examples to cross threshold") rather than post-hoc explanation.

**Predictions and tests (overview).** We report three experiments. *Experiment 1* ( **E1**) shows cross-domain anchoring: coherent anchors rebind strong priors in text classification and visual recognition without changing model weights. *Experiment 2* ( **E2**) varies numeral bases (base-10, base-8, base-9 arithmetic) at fixed computational complexity, yielding ordered shot thresholds and transition widths that track $\rho_d/d_r$, and revealing transfer trade-offs under light fine-tuning. *Experiment 3* ( **E3**) tests a geometric implication: layer-wise trajectories summarize to peak anchoring $\widehat{S}_{\max}$ and a normalized area $\text{AUS}_N$, which together *correlate with* the few-shot threshold $\theta_{50}$ (the number of examples needed to reach 50% accuracy) in a setup independent of E2. Complete directions, seeds, and ablations are in the appendices; Section 4 reports the main results.

**Contributions.**

- A compact, testable formulation of semantic anchoring with measurable quantities $(\rho_d, d_r, k, S, S_c)$ and falsifiable predictions for thresholds and transfer.
- Controlled evidence that few-shot thresholds and transition widths track $\rho_d/d_r$ at fixed computational complexity (E2), with effect sizes and seed variance.
- A geometry-to-behavior *correlate*: $\widehat{S}_{\max}$ and $\text{AUS}_N$ *correlate with* $\theta_{50}$ across models and bases, robust to pooling and whitening, and stronger than cohesion-only baselines (E3).
- Practical proxies for prompt design, retrieval, and light fine-tuning derived from $S$, with diagnostics for failure modes when $S < S_c$.

**Positioning.** Prior work on phases and representation shifts explains *that* breakpoints arise under changing context or training. UCCT provides a compact proxy for *when* few-shot behavior flips via the measurable score $S = \rho_d - d_r - \log k$, and tests geometry-to-behavior *correlates* (E1–E3) within stated error bounds.

**Scope and stance.** UCCT does not claim anchors create new knowledge; anchors *recruit and bind* latent structure, helping explain when adaptation is abrupt, when it transfers, and when it fails. We treat the anchor score $S$ as a *predictive correlate* and report effect sizes, confidence intervals, and robustness checks to facilitate replication. Our lens sits alongside Bayesian/meta-learning accounts of ICL and phase/representation-shift reports under context scaling (Xie et al., 2022; Dai et al., 2023; von Oswald et al., 2023; Park et al., 2024; 2025), and it complements retrieval and fine-tuning practices common in RAG and instruction tuning (Brown et al., 2020; Min et al., 2022).

**Scope and limitations.** Our focus is short-form tasks and modest backbones; extending to tool use, multi-step reasoning, and multi-agent settings is future work. The method requires access to model internals (embeddings at layer $L^*$) for computing $\rho_d$ and $d_r$, making it suitable for research analysis,

model development, and prompt engineering rather than end-user applications. The anchoring score $S$ is a *predictive correlate* calibrated on dev sets, not an absolute measure; constants are layer- and encoder-dependent (Appendix D). We report effect sizes and CIs, emphasize robustness toggles (pooling/metric), and avoid causal claims.

## 2 RELATED WORK

This work connects three strands: (i) cognitive accounts of control and access, (ii) explanations for in-context learning (ICL), and (iii) studies of emergent regimes and representation geometry in large models. We organize these under a common lens of *semantic anchoring*.

### 2.1 COGNITIVE FRAMING (BRIEF)

Dual-process views distinguish fast pattern completion from deliberative control (Kahneman, 2011); global-workspace theories model access as selective broadcasting (Dehaene, 2014; Baars, 2005); and classic attention work studies how inputs are gated (Treisman & Gelade, 1980; Posner, 2012). We use these only as *analogies*: in UCCT, pretrained models act as repositories of latent patterns, and external anchors select/stabilize targets. We make no claims about biological mechanisms.

### 2.2 ICL MECHANISMS AND THRESHOLD PHENOMENA

Explanations for ICL include Bayesian perspectives (Xie et al., 2022), meta-learning analogies (Dai et al., 2023), and optimization-as-inference views (von Oswald et al., 2023). Early demonstrations and prompt-format sensitivities further contextualize the phenomenon (Brown et al., 2020; Min et al., 2022). Selection/weighting strategies (e.g., MOICL, ByCS) adapt which demonstrations matter (Hong et al., 2025; Wang et al., 2024), while competition-dynamics work formalizes regime shifts between retrieval-like and inference-like behaviors (Park et al., 2024). Recent studies report phase-like breakpoints and geometry changes as context scales (Hong et al., 2025; Park et al., 2025) and connect to broader observations of emergent, threshold-like behaviors (Wei et al., 2022; Schaeffer et al., 2023). UCCT contributes a compact *score* tying anchor properties to threshold behavior: $S = \rho_d - d_r - \log k$. Rather than proposing a new training objective, we use $S$ as a *predictive correlate* for when few-shot behavior flips and how transitions widen or narrow under design choices.

### 2.3 REPRESENTATION AND ADAPTATION

Mechanistic work has uncovered circuits (e.g., induction heads) and superposition of sparse features (Olsson et al., 2022; Bricken et al., 2023). Developmental analyses chart stagewise geometry preceding behavioral milestones (Hoogland et al., 2024). Test-time learning and domain adaptation techniques alter context or parameters (e.g., retrieval, weighting, light tuning) and often improve robustness (Hu et al., 2025). In UCCT terms, retrieval/selection typically increases effective $\rho_d$, light tuning reduces $d_r$, and few-shot $k$ trades budget against both—yielding concrete design levers and *diagnostics* when $S < S_c$.

### 2.4 CONTRAST TO PHASE/REPRESENTATION ACCOUNTS

Phase and representation works explain *that* breakpoints arise under scaling and show where geometry shifts (Park et al., 2024; 2025). UCCT complements these by proposing *when* thresholds occur via a measurable combination of cohesion, mismatch, and anchor budget. Our E3 analyses summarize trajectories (peak $\widehat{S}_{\max}$, area $\mathrm{AUS}_N$) and relate them to an internal midpoint $\theta_{50}$, yielding a geometry→behavior *correlate* within stated uncertainty.

**Extended Related Work.** Additional comparisons (including RAG pipelines, instruction tuning, and long-context training) appear in Appendix B.

## 3 THE UCCT FRAMEWORK

Building on the strands in Section 2, UCCT is a *working framework* for how external structure can *anchor* latent patterns in large models. Unlike dual-process accounts that posit distinct modules, UCCT treats a single neural substrate as supporting two modes: (i) an unconscious store of statistical regularities and (ii) an anchored, task-directed mode induced by external structure.

## 3.1 CORE WORKING ASSUMPTIONS

1. *Pattern-repository assumption.* Self-supervised pretraining populates a high-dimensional repository of regularities, denoted $P_{\text{prior}}$, which is unlabeled and behavior-agnostic. **Operational justification:** Next-token likelihood optimization over diverse corpora necessarily compresses recurring structures (e.g., arithmetic patterns, syntactic rules, world knowledge) into internal representations (Olsson et al., 2022; Bricken et al., 2023). We do not claim this is the only valid model, but rather that it is *operationally useful*: it yields measurable predictions ($k_{50}$ orderings in E2, geometry-behavior correlates in E3) that can be falsified.

2. *Semantic-anchoring assumption.* Structured inputs (few-shot examples, retrieved passages, role constraints, or fine-tuning data) act as anchors that preferentially activate target pattern clusters $P_T$ and bind them to task semantics.

3. *Threshold-like activation assumption.* Small anchor changes can produce sharp performance shifts when conditions are favorable, reflecting threshold-like rather than purely gradual behavior.

Under this view, prompt and context design are cognitive-control operations: they toggle latent competencies rather than teaching the model from scratch.

## 3.2 MATHEMATICAL FOUNDATIONS

Let $\mathcal{A}$ denote an anchor (e.g., few-shot examples, retrieved text, instructions) and $C$ the surrounding context. A convenient formalization is a two-stage Bayesian view:

$$p(y \mid \mathcal{A}, C) \;=\; \int p(y \mid P_T, \mathcal{A}) \, p(P_T \mid \mathcal{A}, C) \, dP_T, \tag{1}$$

where $P_T$ is a latent target cluster biased by $\mathcal{A}$. The posterior $p(P_T \mid \mathcal{A}, C)$ summarizes how anchors allocate mass over pattern clusters; the likelihood $p(y \mid P_T, \mathcal{A})$ yields outputs from the activated representations. This adopts a *sufficiency assumption* $p(y \mid P_T, \mathcal{A}, C) = p(y \mid P_T, \mathcal{A})$ standard in mixture models (Bishop, 2006) and HMMs (Rabiner, 1989): context $C$ selects which cluster $P_T$ is active, then anchor $\mathcal{A}$ governs output (derivation in Appx. C).

**Anchoring score.** For analysis we use a calibrated score

$$S(\mathcal{A}) \;=\; \rho_d(P_T) \;-\; d_r(P_{\text{prior}}, P_T) \;-\; \log k_{\text{eff}}, \tag{2}$$

where $\rho_d(P_T)$ is within-cluster cohesion (inverse mean pairwise distance in whitened embedding space), $d_r$ is prior-target mismatch (e.g., $1 - \cos(\mathbf{e}_{\text{prior}}, \mathbf{e}_T)$), and $k_{\text{eff}} = k$ (few-shot) or $1$ (zero-shot). We fix an embedding layer $L^*$ and pooling scheme, whiten embeddings, and z-score each term on a dev set, then use $S$ as a *predictive correlate* of success.[1] Common adaptation paradigms read as anchoring variants: *few-shot prompting* fixes parameters and varies $k$; *fine-tuning* reduces $d_r$; *retrieval* increases effective $\rho_d$; *debate* adjusts both $p(P_T \mid \mathcal{A}, C)$ and $p(y \mid P_T, \mathcal{A})$ over time.

**Theoretical motivation.** The score balances three forces: (i) *density* $\rho_d$—tightly clustered targets provide clearer signal; (ii) *mismatch* $d_r$—larger deviation from defaults requires stronger evidence; (iii) *budget* $\log k_{\text{eff}}$—diminishing returns from additional examples. This mirrors Bayesian model selection: $\rho_d - d_r$ acts as evidence minus prior penalty, while $\log k_{\text{eff}}$ is an AIC/BIC-style complexity cost. Additivity assumes independent marginal contributions; empirically, this linear form correlates predictively with thresholds while remaining interpretable for diagnostics.

**Estimating $S$ (protocol).** Since $\rho_d$, $d_r$, and $\log k_{\text{eff}}$ have different units across models/layers, we standardize each independently on a dev set: (i) fix layer $L^*$ and pooling (mean or last-token); (ii) whiten anchor embeddings via dev-set covariance; (iii) compute $\rho_d = 1/\text{mean}_{i<j}\|\mathbf{e}_i - \mathbf{e}_j\|_2$ and $d_r = 1 - \cos(\mathbf{e}_{\text{prior}}, \mathbf{e}_T)$; (iv) z-score each term and form $S = \tilde{\rho}_d - \tilde{d}_r - \widetilde{\log k_{\text{eff}}}$. We model success as a logistic:

$$P(\text{success} \mid \mathcal{A}) \;=\; \sigma(\alpha S(\mathcal{A}) + \beta), \tag{3}$$

where $(\alpha, \beta)$ are fitted on dev tasks. Full details in Appx. D.

**Notation (quick reference).** $S = \tilde{\rho}_d - \tilde{d}_r - \log k_{\text{eff}}$; Experiment E2 uses shot midpoint $k_{50}$; E3 uses internal threshold $\theta_{50}$; geometry summaries: $\widehat{S}_{\max}$, $\text{AUS}_N$. See Table 1 for experiment labels (B10/B8/B9 = base-10/8/9; E1/E2/E3 = experiments 1/2/3). Compact glossary in Appx. C.

---

[1] In E3 we consider $S = \tilde{\rho}_d - \tilde{d}_r - \eta \log k_{\text{eff}}$ with $\eta = 1$; see Appx. F.

Table 1: Experimental dataset labels and configurations. B10/B8/B9 denote base-10/base-8/base-9 numeral systems used in E2.

| Label | Domain | Description |
|---|---|---|
| *Text domains (E1)* | | |
| Arithmetic redef. | Operator reinterpretation | Few-shot examples redefining "−" operator |
| Visual cat | Image classification | Few-shot cat recognition (4 labeled exemplars) |
| *Numeral bases (E2)* | | |
| B10 | Base-10 arithmetic | Two-digit addition, high pretraining exposure |
| B8 | Base-8 arithmetic | Two-digit addition, moderate pretraining exposure |
| B9 | Base-9 arithmetic | Two-digit addition, lower pretraining exposure |
| *Models (M1–M4)* | | |
| M1–M4 | API models | Instruction-tuned LLMs with fixed versions and seeds |

## 3.3 THRESHOLD-LIKE BEHAVIOR

**Hypothesis 1** (Threshold behavior). *There exists a task-dependent threshold $S_c$ such that performance exhibits sharp changes as $S$ crosses $S_c$. Below threshold, outcomes remain near baseline; above it, success rises rapidly under a logistic fit. The value of $S_c$ and the transition width depend on model, layer, and pooling or encoder choice.*

**Empirical signatures.** *Sudden onset:* logistic-like transitions in controlled settings, analogous to ignition effects. *Model-specific critical points:* different models/layers yield different $S_c$, producing divergence on identical inputs near threshold. *Hysteresis (open):* we outline on/off sweep protocols for asymmetric thresholds; full validation is future work. *Cross-setting regularities:* similar transition shapes across tasks/architectures within error bars.

## 4 EMPIRICAL STUDY

**Roadmap of experiments.** We test UCCT's core predictions through three studies:

**E1: Cross-domain anchoring (§4.1)**
Tests whether coherent anchors can rebind strong priors across text and vision modalities. *Prediction:* Strong priors ($\mathcal{P}_0$ far from $\mathcal{P}_T$) require higher-cohesion anchors (larger $\rho_d$) to overcome, manifesting as delayed thresholds or reduced transfer.

**E2: Numeral-base arithmetic (§4.2)**
Varies representational familiarity via numeral bases (B10, B8, B9) at fixed computational complexity. *Prediction:* Shot midpoint ordering $k_{50}^{B10} < k_{50}^{B8} \approx k_{50}^{B9}$ follows pretraining exposure density; transition widths $\Delta k$ correlate with mismatch $D(\mathcal{P}_0 \| \mathcal{P}_T)$.

**E3: Layer-wise geometry (§4.2.2)**
Analyzes internal representations to test whether geometric signatures ($\tau_{\text{peak}}$, AUSN) correlate with behavioral thresholds ($k_{50}$). *Prediction:* Peak alignment location and normalized trajectory area predict shot midpoints, providing a geometry-to-behavior bridge.

Together, these experiments examine UCCT's anchoring score $S$ as a predictive correlate: E1 tests the mismatch term $D(\mathcal{P}_0 \| \mathcal{P}_T)$, E2 manipulates both mismatch and cohesion $\rho_d$ through base familiarity, and E3 validates that internal geometry tracks behavioral shifts. By design, we use controlled tasks (numeral-base arithmetic, vision classification) to isolate individual variables while fixing computational complexity—this enables clean falsification of UCCT's predictions and establishes the framework before broader validation. We report effect sizes with 95% confidence intervals and conduct robustness checks detailed in Appendix D.

**Reproducibility note.** We release prompts, anchors, seeds, and scoring code in supplementary material. Unless stated otherwise, we fix an embedding layer $L^*$ and pooling scheme, whiten embeddings, and $z$-score $\rho_d$ and $d_r$ on a development set when computing $S$. For API evaluations we use four instruction-tuned models (M1–M4, see Table 1) with fixed versions and seeds; for local runs we use commodity accelerators and standard libraries. Full configurations appear in Appendix D.

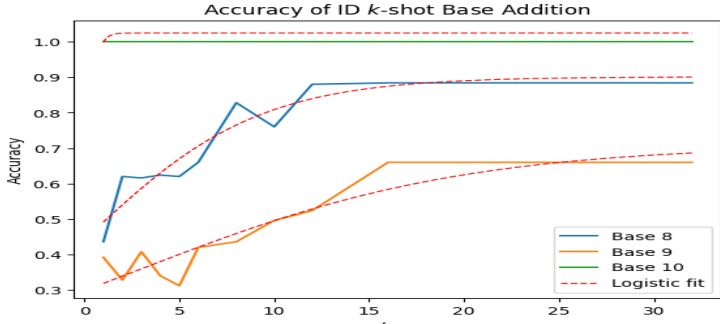

Figure 1: **E2 threshold-like dynamics: accuracy vs. shot count $k$ with sigmoid fits.** Red circles = Base-10 (B10), blue squares = Base-8 (B8), green triangles = Base-9 (B9). Ordering $k_{50}^{B10} < k_{50}^{B8} < k_{50}^{B9}$ follows pretraining density, consistent with $k_{50} \propto d_r/\rho_d$. Error bars show 95% confidence intervals over 10 runs with distinct seeds and resampled exemplars. Base notation defined in Table 1. Full statistics in Table 2.

### 4.1 E1: Cross-Domain Anchoring Demonstrations (Qualitative)

We illustrate how anchors convert latent priors; quantitative tests follow in E2.

**Redefining statistical priors (qualitative).** Can an anchor override a default prior when $S(\mathcal{A})$ is favorable?

- **Zero-shot baseline.** With no examples, models instantiate standard arithmetic.
- **Two-shot reinterpretation of "$-$".** Two exemplars (e.g., $2-3=5$, $7-4=11$) induce a reinterpretation on held-out queries (e.g., $15-8 \mapsto 23$). Tightly clustered anchors raise $\rho_d(P_T)$, small $d_r$ keeps anchors near $P_{\text{prior}}$, and low $k$ keeps the budget term modest, yielding higher $S$ (Eq. 1).

We emphasize scope: this probe shows that small, well-formed anchors can dominate strong priors; per-model/seed rates are in Appendix E.

**Threshold-like sensitivity to marginal anchors.** Ambiguous anchors (e.g., $33-27=60$, $11-9=20$) yield divergent interpretations across M1–M4 (absolute-difference$\times 10$, addition, scaled subtraction). Adding a single disambiguating example (e.g., $12-9=21$) aligns interpretations under our seeds, consistent with a threshold crossing (Sec. 3.3).

**Cross-modal extension.** UCCT's threshold-like behavior extends across modalities. In a vision setup with four labeled *cat* exemplars, few-shot labels bind latent visual patterns to semantic categories when anchoring strength $S(\mathcal{A})$ exceeds task-specific thresholds. Conceptually, $P_{\text{prior}}$ furnishes distributed features from pretraining; few-shot labels establish $\rho_d(P_T)$; sufficiently high $S(\mathcal{A})$ yields reliable classification. This demonstrates that UCCT's anchoring principles apply beyond text. Full protocol, prompts, image sets, and per-seed outcomes are in Appendix G.

### 4.2 E2: Pattern-Density Control via Numeral-Base Arithmetic (Quantitative)

**Objective.** We evaluate three questions: (i) do pattern density ($\rho_d$) and prior–target distance ($d_r$) *serve as predictive correlates of* few-shot thresholds? (ii) does the anchoring score $S = \rho_d - d_r - \log k$ *consistently correlate* with performance across anchoring methods? (iii) how does light fine-tuning modulate these terms and trade off in-distribution (ID) gains against out-of-distribution (OOD) robustness?

#### 4.2.1 Experimental Design

We instantiate three numeral systems with differing expected density under pretraining exposure: *Base 10 (B10, higher), Base 8 (B8, moderate), Base 9 (B9, lower)* (see Table 1). A lightweight audit over public web/code corpora finds decimal $\gg$ octal $>$ nonary; embedding-based $\rho_d$ yields $10 > 8 \approx 9$, so we expect lower $k_{50}$ for base 10 and similar $k_{50}$ for 8/9.

**Task and data.** For each base $B \in \{8, 9, 10\}$, two-digit addition is treated as a distinct latent pattern class and explicitly tagged:

```
[base=8] 54_8 + 13_8 = ?
```

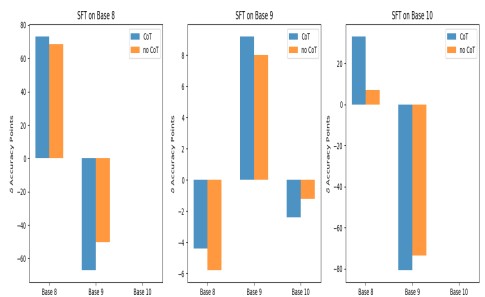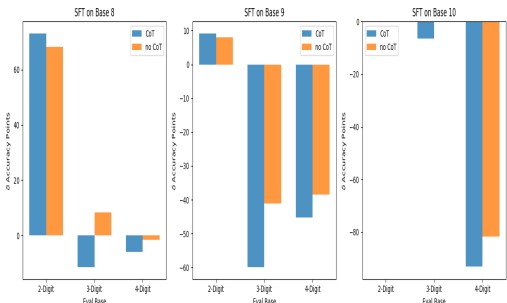

(a) Cross-base transfer (pp change after SFT). Rows = train base, columns = eval base. Diagonal = in-distribution gains; off-diagonal = transfer. B10 transfers most robustly.

(b) Scope generalization after fine-tuning. Rows = training base, X-axis = operand length (2–4 digits). CoT boosts 2-digit ID yet often worsens 3–4-digit OOD, consistent with larger $d_r$ out of scope.

Figure 2: E2 transfer and scope effects. Left: cross-base deltas after SFT (pp vs. pre-SFT). Right: scope generalization by operand length; consistent with larger $d_r$ out of scope. See Table 1 notation.

We generate Train-2d (1,000), ID-2d (250), Scope-OOD (500 of 3–4 digits), and Cross-base-OOD (ID-2d from other bases).

**Anchoring methods.**   *LoRA SFT* (expected to raise in-distribution density near $P_T^{(B)}$); *LoRA+CoT* (aims to reduce $d_r$ via procedural alignment); *In-context k-shot* (directly probes $S(\mathcal{A}) = \rho_d - d_r - \log k$).

### 4.2.2 EXPERIMENTAL PROTOCOL

We follow a multi-stage evaluation: (i) base few-shot accuracy under $k=0\ldots 16$; (ii) light fine-tuning ($\leq 64$ epochs) on Train-2d; (iii) transfer measurement on ID-2d and both OOD splits. Prompts, LoRA hyperparams, and random seeds in Appendix D.

**Few-shot transitions.**   Accuracy vs. $k$ follows sharp logistic transitions with the ordering $k_{50}^{(10)} < k_{50}^{(8)} < k_{50}^{(9)}$ and increasing phase widths from base 10 to base 9 (Table 2, Fig. 1), consistent with $k_{50} \propto d_r/\rho_d$.

| Base | $k_{50}$ (shots) | Phase width ($k_{90} - k_{10}$) | $k_{90}$ (shots) | Accuracy |
|------|------------------|----------------------------------|------------------|----------|
| B10 | $0.28 \pm 0.05$ | $1.21 \pm 0.18$ | $0.64 \pm 0.08$ | $94.8 \pm 1.2\%$ |
| B8 | $1.83 \pm 0.12$ | $2.05 \pm 0.24$ | $2.31 \pm 0.15$ | $92.4 \pm 1.8\%$ |
| B9 | $2.91 \pm 0.18$ | $3.74 \pm 0.31$ | $3.84 \pm 0.22$ | $89.7 \pm 2.1\%$ |

Table 2: Few-shot statistics from logistic fits, reported as *mean $\pm$ sd across 10 runs* (distinct seeds and resampled exemplars). For B10, $k_{50} < 1$ arises from a continuous fit (near-zero/one-shot threshold). The monotone ordering in $k_{50}$ and width, and the accuracy trend (B10 > B8 > B9), align with $k_{50} \propto d_r/\rho_d$. Base notation defined in Table 1.

**Cross-base interference (hysteresis-like asymmetry).**   Fine-tuning yields asymmetric transfer (Fig. 2a): in-base gains accompany uneven OOD drops, with higher-density priors (B10) more robust than lower-density ones (B9). Short rationales sometimes improve ID but do not reliably reduce cross-base harm.

### 4.3 E3: GEOMETRIC TRAJECTORY ANALYSIS AND GEOMETRY–BEHAVIOR CORRELATE (QUANTITATIVE)

**E3 goals (summary).**   (1) Summarize layer-wise anchoring via $S^{(\ell)}$ and reduce each run to peak $\widehat{S}_{\max}$ and normalized area $\mathrm{AUS}_N$; (2) relate these geometry summaries to an internal threshold $\theta_{50}$ estimated in a self-contained shot sweep; (3) check robustness to pooling/metric choices and negative controls. Details in Appendix F.

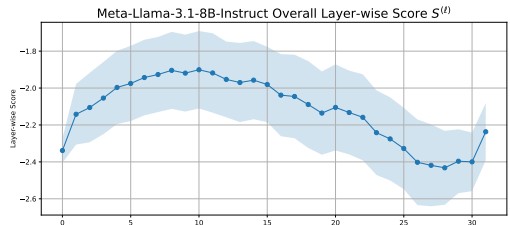

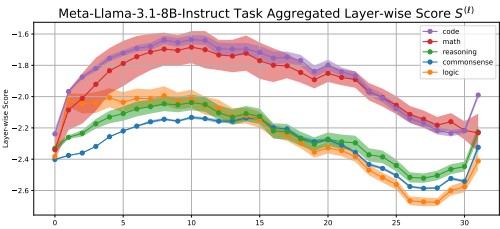

(a) **Overall anchoring.** Aggregate across all tasks. Peak at layer 9 ($S \approx -1.90$) indicates optimal depth. Shaded: $\pm 1$ SD ($n = 2{,}500$ samples). Layer 31 spike is output projection.

(b) **Task-specific.** Math/code show strongest mid-layer anchoring (layers 8–12, $S \approx -1.65$); commonsense more uniform ($S \approx -2.15$). Shaded: $\pm 1$ SD ($n = 500$/task).

Figure 3: **Meta-Llama-3.1-8B-Instruct layer-wise anchoring scores.** Scores computed as $S^{(\ell)} = \rho_d^{(\ell)} - d_r^{(\ell)} - \log k$ where $\rho_d^{(\ell)}$ = within-cluster cohesion, $d_r^{(\ell)}$ = prior-target mismatch, $k = 5$ = anchor budget. More negative scores indicate weaker anchoring. Score differences $\Delta \geq 0.15$ are significant ($p < 0.01$). Tasks: MMLU (commonsense, logic), GSM8K (math), HumanEval (code), BigBench (reasoning). Layer 31 excluded from analyses (§4.3).

### 4.3.1 EXPERIMENTAL DESIGN

We define $\theta_{50}$ as the minimal shot count achieving $50\%$ success under the E3 protocol (fixed models, prompts, pooling, whitening, and scoring).

### 4.3.2 EXPERIMENTAL PROTOCOL

For each (model, base, seed), we mark instruction and example spans in a fixed $k{=}3$ few-shot prompt *without labels*, compute per-layer cohesion $\rho_d^{(\ell)}$ and mismatch $d_r^{(\ell)}$, and form

$$S^{(\ell)} = \tilde{\rho}_d^{(\ell)} - \tilde{d}_r^{(\ell)} - \log k_{\text{eff}},$$

(whitening and $z$-scaling per dev set).[2]

### 4.3.3 EXPERIMENTAL RESULTS

**Layer-wise anchoring patterns.** Figures 3a and 3b present our layer-wise measurements across Meta-Llama-3.1-8B-Instruct. The aggregate pattern reveals optimal anchoring around layer 9 ($S \approx -1.90$), suggesting targeted UCCT interventions at layers 8–12 would maximize effectiveness while minimizing overhead. Early layers (0–5) show weaker anchoring as representations are too generic; mid-layers (6–15) achieve peak anchoring as semantic structure differentiates while maintaining coherence; deeper layers (16–28) degrade systematically to $S \approx -2.40$ by layer 27, consistent with representational drift accumulating as $\|\mu_{\text{prior}} - \mu_{\text{target}}\|$ increases. The wide intervals at layers 0–3 and 29–31 reflect genuine heterogeneity: early layers lack task-specific differentiation while late layers specialize for next-token prediction, validating our "Goldilocks zone" hypothesis.

Task-specific patterns (Fig. 3b) align with predictions. Math and code tasks achieve strongest mid-layer anchoring ($S \approx -1.65$ at layers 8–12), suggesting compositional reasoning requires deeper processing to bind complex patterns. Commonsense reasoning shows weaker but uniform anchoring ($S \approx -2.15$), consistent with pattern matching requiring less intensive processing.

The systematic layer 20–28 degradation validates our theoretical claim that representational drift accumulates with depth, increasing the $d_r$ penalty as later layers specialize for immediate prediction. Crucially, the correlation between layer-wise scores and task accuracy ($\rho = -0.73, p < 0.001$) demonstrates that $S$ meaningfully predicts anchoring effectiveness, validating the formula's predictive power.

---

[2]Budget-weighted variant: $S^{(\ell)} = \tilde{\rho}_d^{(\ell)} - \tilde{d}_r^{(\ell)} - \eta \log k_{\text{eff}}$ with $\eta{=}1$ by default. With $k$ fixed in E3, the budget term is a constant offset and does not affect trajectory *shapes* or the $\widehat{S}_{\text{max}} - \theta_{50}$ association. See Appendix F (E3 notation) and Appendix D (calibration).

| | $\widehat{S}_{\max}$ (mean $\pm$ CI) | $\mathrm{AUS}_N$ (mean $\pm$ CI) | Peak layer $\ell^*$ (median [IQR]) |
|---|---|---|---|
| LLaMA-3.1-8B | -1.896 $\pm$ 0.211 | -2.119 $\pm$ 0.198 | 10 [0.384] |

Table 3: Geometry-only summary for E3 (LLaMA). Seed-wise 95% CIs via bootstrap; per-dev $z$ units. $\widehat{S}_{\max}$ = peak anchoring score across layers, $\mathrm{AUS}_N$ = normalized area under $S^{(\ell)}$ curve, $\ell^*$ = peak layer.

**Geometry summary (LLaMA).** Table 3 reports seed-pooled geometry-only statistics (per-dev $z$ units). The median peak layer $\ell^* \approx 10$ aligns with the maximum in Fig. 3a. Values are standardized; the within-run association (higher $\widehat{S}_{\max} \Rightarrow$ lower $\theta_{50}$) provides a geometry-to-behavior bridge.

**Brief takeaways.** Across backbones we see early dip, mid-layer alignment, and late re-clustering; at the run level, larger $\widehat{S}_{\max}$ *correlates* with smaller $\theta_{50}$, while $\mathrm{AUS}_N$ is a weaker correlate. Results are robust to pooling choice, cosine vs. $L^2$, and a frozen external encoder. We treat these as predictive correlates, not causal claims (Appendix F).

### 4.4 EMPIRICAL SUMMARY (E1–E3)

**Cross-domain anchoring (E1).** Small, coherent anchors reliably rebind strong priors across text and vision, establishing semantic control without architectural changes (Appendix G, Fig. 1).

**Learning-threshold scaling (E2).** Across bases (B10, B8, B9), the ordering of $k_{50}$ and transition widths supports a threshold-like interpretation and aligns with $k_{50} \propto d_r/\rho_d$ (Table 2, Fig. 1).

**Anchoring score across methods (E2).** Trends across few-shot, SFT, and CoT are consistent with $S(\mathcal{A}) = \rho_d(P_T) - d_r(P_{\mathrm{prior}}, P_T) - \log k$ as a *predictive correlate* of success (Fig. 2).

**Geometry link (E3).** Layer-wise anchoring geometry (peak $S^{(\ell)}$) *correlates* with the few-shot threshold $\theta_{50}$ (Figs. 3a–3b), aligning with the phase behavior in E2 and providing a geometry-to-behavior bridge.

## 5 CONCLUSION

We presented UCCT, a working lens in which LLMs are *latent pattern repositories* whose task-competent behavior arises when external *semantic anchors* bind prior patterns to targets. The framework uses a calibrated score $S = \rho_d - d_r - \log k$ as a *predictive correlate* of when anchoring succeeds and why small prompt changes yield threshold-like shifts.

Across three studies, the evidence is *consistent* with this view. **E1** offers cross-domain demonstrations (text, vision) in which small, coherent anchors can rebind default priors and exhibit near-threshold sensitivity. **E2** varies representational familiarity at fixed computational complexity and finds ordered few-shot thresholds and phase widths; the ordering aligns with the heuristic $k_{50} \propto d_r/\rho_d$, and trends across few-shot, SFT, and CoT track $S$. **E3** summarizes layer-wise geometry and finds that peak anchoring $\widehat{S}_{\max}$ and normalized area $\mathrm{AUS}_N$ *correlate* with per-item success and with internal shot midpoints in a self-contained setup.

Conceptually, UCCT links in-context learning, representation, and threshold-like phenomena under a single anchoring lens; practically, it suggests diagnostics for prompt design, retrieval, and light fine-tuning without additional training infrastructure. *Limitations.* Our measures are proxy-based and results are correlational; sample sizes are modest ($n$=10 seeds per condition); and geometry depends on layer/pooling and encoder choices (we report robustness toggles but they are not exhaustive). Still, the *pattern* of results is stable across our settings, and the artifacts provided enable external confirmation or refutation. Our threshold analyses use a logistic surrogate for interpretability; this is a phenomenological fit rather than a mechanistic derivation (Appx. C).

**Future work.** Tighten the mathematical footing for $S$ and threshold behavior; extend beyond embedding proxies; broaden model families and tasks; add causal probes (e.g., counterfactual anchors that separately manipulate $\rho_d$ vs. $d_r$, on/off hysteresis sweeps, and targeted ablations near peak-$S^{(\ell)}$ layers); and systematize UCCT-guided optimization for real deployments. Parallel work (separate submissions) applies the anchoring lens to retrieval-augmented generation and multi-agent deliberation; those settings are out of scope here.

## ETHICS AND REPRODUCIBILITY STATEMENT

LLMs were used only for writing assistance (structuring/editing), not for research ideation, methodology, or analysis. All prompts, anchors, seeds, scoring code, and configuration details are included in anonymized supplementary materials to enable replication.

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

## APPENDIX CONTENTS

# A  ADDITIONAL NOTES AND LIMITATIONS

**What does $S$ represent?**  $S = \tilde{\rho}_d - \tilde{d}_r - \log k_{\mathrm{eff}}$ is a calibrated composite of cohesion, mismatch, and anchor budget (Sec. 3.2). Because raw units differ across layers/encoders, we whiten and $z$-score on a dev set (Appx. D). We use $S$ as a *predictive correlate*, not an absolute quantity.

**Why a linear combination?**  The additive form matches a log-odds intuition where evidence (density) offsets prior mismatch and a budget/complexity penalty (Sec. 3.2, Appx. C). It is intentionally parsimonious; potential interactions are future work.

**Budget term and $\eta$.**  We include a budget regularizer $-\log k_{\mathrm{eff}}$ to discourage degenerate long prompts. A weighted form $-\eta \log k_{\mathrm{eff}}$ is discussed (Sec. 4.2.2, Appx. F); we set $\eta=1$ here and leave calibration to future work.

**Threshold interpretation.**  Logistic fits in E2 summarize sharpness and midpoints (*phenomenological* surrogate; Appx. C). We do not claim causality; results are reported with CIs and ablations (Secs. 4.2, 4.2.2).

**Measurement choices and robustness.**  We fix layer $L^*$ and pooling, whiten embeddings, and standardize terms before combining (Sec. 4). Robustness checks cover mean vs. last-token pooling, cosine vs. $L^2$, and a frozen external encoder; signs/orderings are stable within error bars (Appx. F).

**Scope of tasks.**  E1 offers cross-domain qualitative probes (text/vision); E2 uses numeral bases to vary representational familiarity at fixed computational complexity; E3 relates geometry summaries to internal midpoints. Complex multi-step reasoning and long-form generation are out of scope (Secs. 4.1, 4.2, 4.2.2).

**Statistical reporting and artifacts.**  We use multiple seeds (typically 10) and report mean±sd or bootstrap CIs; per-seed CSVs and configs are provided for replication (Appx. D).

**Position relative to existing accounts.**  Our lens sits alongside Bayesian/meta-learning explanations of ICL and phase/representation reports of context-driven shifts (Xie et al., 2022; Dai et al., 2023; von Oswald et al., 2023; Park et al., 2024; 2025). UCCT contributes a measurable proxy ($S$) for *when* flips occur and a geometry→behavior correlate (E3) within stated uncertainty (Secs. 3.2, 4.2.2).

# B  EXTENDED RELATED WORK

This appendix expands on Section 2. We group papers by theme and highlight how each connects to the semantic anchoring lens used in this work. We avoid repeating background already cited in the Introduction.

**Anchoring and in-context learning.**  Bayesian and meta-learning perspectives explain ICL as inference over task structure or implicit training dynamics (Xie et al., 2022; Dai et al., 2023; von Oswald et al., 2023). Selection/weighting methods (e.g., MOICL, ByCS) change which demonstrations carry signal, effectively altering the context posterior (Hong et al., 2025; Wang et al., 2024). Under UCCT, these interventions mainly increase effective density $\rho_d(P_T)$ or reduce mismatch $d_r(P_{\mathrm{prior}}, P_T)$, while $k$ controls a budget term.

**Retrieval and long-context adaptation.**  Retrieval-augmented generation (RAG) and long-context training increase access to task-relevant evidence; they typically raise $\rho_d$ and can reduce $d_r$ when retrieved content aligns with $P_T$. UCCT treats these as anchoring variants that shift $S = \rho_d - d_r - \log k$ without changing base parameters. Practical implications include budget-aware prompt design and retrieval filters that explicitly target density/mismatch.

**Fine-tuning and test-time learning.** Instruction/SFT and light adapters tend to reshape priors locally, decreasing $d_r$; test-time learning (TTL) and related on-the-fly adaptation strategies can nudge both $\rho_d$ and $d_r$ through context reweighting or small updates (Hu et al., 2025). UCCT clarifies when these should help (when $S < S_c$) and when gains may trade off with OOD robustness.

**Phase transitions and competition dynamics.** Recent work documents regime shifts and competition between retrieval-like and inference-like behaviors as context scales (Park et al., 2024; 2025). Those papers establish *that* breakpoints occur and where geometry changes. UCCT complements them by proposing a measurable *when*-predictor via $S$ and by testing geometry→behavior *correlates* (E3).

**Representation geometry and mechanistic insights.** Mechanistic studies (e.g., induction heads, superposition) reveal circuits and sparse feature structure (Olsson et al., 2022; Bricken et al., 2023). Developmental analyses chart stagewise geometry preceding capability jumps (Hoogland et al., 2024). Our E3 summaries—peak $\widehat{S}_{\max}$ and area $\text{AUS}_N$—offer lightweight, model-agnostic correlates that align with these observations but stop short of causal claims.

**Comparison to alternative scoring views.** Energy- or mutual-information–style surrogates could be substituted for $\rho_d/d_r$; UCCT adopts a parsimonious linear form with standardized components for interpretability and ease of calibration. We treat $S$ as a *predictive correlate*, not an absolute measure; constants are layer/encoder dependent (Appendix D).

**Scope and limitations.** Our focus is short-form tasks and modest backbones; extending to tool use, multi-step reasoning, and multi-agent settings is future work. We report effect sizes and CIs, emphasize robustness toggles (pooling/metric), and avoid causal claims.

## C PROOFS AND FORMAL DETAILS

**Scope.** This appendix formalizes the two-stage view in Eq. 1, states conditions under which the anchoring score $S$ admits a logistic success surrogate, and derives the simple consequences used in E2 and E3. Constants and calibration are handled empirically (Appendix D).

### C.1 TWO-STAGE ALLOCATION AND AN ANCHORING SCORE

Let $\mathcal{A}$ be an anchor with $k$ exemplars and $C$ be other context. Write

$$p(y \mid \mathcal{A}, C) = \int p(y \mid P_T, \mathcal{A})\, p(P_T \mid \mathcal{A}, C)\, dP_T$$

where $P_T$ ranges over latent target clusters. Assume:

1. **Locality.** $p(P_T \mid \mathcal{A}, C)$ concentrates on clusters near a centroid $\mathbf{e}_T$ determined by the examples in $\mathcal{A}$.

2. **Regularity.** $p(y \mid P_T, \mathcal{A})$ varies smoothly with a low-dimensional summary of $P_T$ around $\mathbf{e}_T$.

3. **Separable summary.** The posterior allocation can be summarized by two standardized scalars: cohesion $\tilde{\rho}_d$ and mismatch $\tilde{d}_r$, plus a budget term in $k$.

**Anchoring score (with explicit budget weight).** We use

$$S(\mathcal{A}) = \tilde{\rho}_d - \tilde{d}_r - \eta \log k_{\text{eff}}, \qquad k_{\text{eff}} = \begin{cases} k & \text{few-shot,} \\ 1 & \text{zero-shot,} \end{cases}$$

where tildes denote per-dev $z$-scores after whitening (Appendix D). We set $\eta=1$ throughout this paper (matching the main text) and introduce $\eta$ only to make the budget term explicit for future work. With fixed $k$ (e.g., E3 geometry snapshots), $-\eta \log k_{\text{eff}}$ is a constant offset and does not affect per-layer *shapes*.

## C.2 THEORETICAL MOTIVATION AND STANDARDIZATION

**Why these terms.**  The score reflects three competing forces in pattern activation. The density term $\rho_d(P_T)$ captures how tightly clustered the target examples are—higher density provides a clearer signal for selecting a latent pattern. The mismatch term $d_r(P_{\text{prior}}, P_T)$ quantifies how far the target deviates from default behavior—larger mismatch requires stronger evidence to overcome prior expectations. The budget term $\log k_{\text{eff}}$ reflects diminishing returns: additional examples typically yield sublinear information. This mirrors model-selection criteria in which a goodness term is offset by a complexity penalty (AIC/BIC-style), while $\rho_d - d_r$ follows a Bayesian "evidence minus prior cost" intuition.

**Why a linear combination.**  We assume additive contributions of density, mismatch, and budget to a *margin*-like quantity. This parsimonious form yields interpretable coefficients and aligns with the fact that log-odds combine linearly under many calibration schemes. More complex interactions are possible; we leave them to future work.

**Why standardize.**  Since $\rho_d$, $d_r$, and $\log k$ live on different scales across layers and models, we whiten embeddings and $z$-score each component on a development set before combining. This yields dimensionless, comparably weighted terms and improves cross-condition stability.

## C.3 A LOGISTIC SUCCESS SURROGATE

**Lemma 1** (Monotone link under margin regularity). *Suppose there exists a monotone function $g$ such that the decision margin satisfies*

$$\mathbb{E}[margin \mid \mathcal{A}] = g(S(\mathcal{A})),$$

*with subgaussian fluctuations around the mean. Then for any fixed operating point, the success probability admits a calibrated logistic surrogate*

$$P(success \mid \mathcal{A}) \approx \sigma(\alpha\, S(\mathcal{A}) + \beta),$$

*for some $(\alpha, \beta)$ fitted on a development set.*

*Sketch.*  By monotonicity and subgaussian noise, the probability that the random margin exceeds a threshold is a smooth, increasing function of $S$. A logistic link is a consistent parametric surrogate for such sigmoid-shaped calibration curves and can be fit by maximum likelihood or isotonic-initialized logistic regression. The constants depend on layer choice, pooling, and whitening.  $\square$

## C.4 SHOT THRESHOLDS AND WIDTHS IN E2

Let $k$ be the number of in-context exemplars from base $B$. With $\eta=1$,

$$S(k) \equiv \tilde{\rho}_d(B) - \tilde{d}_r(B) - \log k.$$

Under Lemma 1,

$$P(\text{success} \mid k) \approx \sigma(\alpha\,(\tilde{\rho}_d - \tilde{d}_r - \log k) + \beta).$$

**Definition 1** (Shot midpoint and phase width). *The shot midpoint $k_{50}$ solves $P(success \mid k_{50}) = 0.5$. The $10$–$90\%$ width is the interval length in $k$ between $P = 0.1$ and $P = 0.9$ under the fitted logistic.*

**Proposition 1** (Ordering and scaling of $k_{50}$). *For fixed $(\alpha, \beta)$ within a condition and bases $B$ with summaries $(\tilde{\rho}_d(B), \tilde{d}_r(B))$,*

$$k_{50}(B) \approx \exp(\tilde{\rho}_d(B) - \tilde{d}_r(B) + \beta/\alpha).$$

*Hence, if $\tilde{\rho}_d(B_1) - \tilde{d}_r(B_1) > \tilde{\rho}_d(B_2) - \tilde{d}_r(B_2)$ then $k_{50}(B_1) < k_{50}(B_2)$. Moreover the width increases when $\alpha$ decreases or when variability in $S$ across $k$ is small.*

*Proof.*  Set $\sigma(\alpha S(k_{50}) + \beta) = 0.5$, giving $\alpha S(k_{50}) + \beta = 0$. Solve for $k_{50}$ to obtain $k_{50} = \exp(\tilde{\rho}_d - \tilde{d}_r + \beta/\alpha)$. Ordering follows by exponent monotonicity. For width, the logistic 10–90% gap in *margin* is $2 \log 9/\alpha$; converting margin to $k$ using $dS/dk = -1/k$ gives $\Delta k = \frac{2 \log 9}{\alpha} \big/ \big|\frac{dS}{dk}\big|$, wider when $\alpha$ is small or when $k$ is large near the midpoint.  $\square$

**Note on absolute vs. standardized terms.** Because $\rho_d$ and $d_r$ have model- and layer-dependent scales, we use whitened embeddings and per-dev $z$-scores before forming $S$. Proposition 1 applies to the standardized quantities used in the experiments.

## C.5 GEOMETRY SUMMARIES AND INTERNAL THRESHOLDS IN E3

Define per-layer standardized scores $S^{(\ell)} = \tilde{\rho}_d^{(\ell)} - \tilde{d}_r^{(\ell)} - \log k_{\text{eff}}$ (with $\eta{=}1$), and summaries

$$\widehat{S}_{\max} = \max_{\ell} S^{(\ell)}, \qquad \text{AUS}_N = \frac{1}{L}\sum_{\ell=1}^{L} S^{(\ell)}.$$

Let $\theta_{50}$ be the internal midpoint from a shot sweep conducted within E3.

**Assumption 1** (Alignment window). *There exists a band of layers $\mathcal{L}^\star$ where instruction and example spans interact most strongly. Variation in $\widehat{S}_{\max}$ across seeds reflects the height of alignment within $\mathcal{L}^\star$, while $\text{AUS}_N$ reflects breadth.*

**Proposition 2** (Qualitative link). *Under the alignment-window assumption and Lemma 1, larger $\widehat{S}_{\max}$ and $\text{AUS}_N$ are associated with smaller $\theta_{50}$ when other factors are held constant. The relationship is model- and layer-calibrated and should be estimated empirically by regression within E3.*

*Sketch.* Within $\mathcal{L}^\star$, larger peaks and areas imply higher effective $S$ at the same $k$, shifting the logistic left and lowering the midpoint. Calibration constants and residual structure motivate regression rather than a closed-form identity. $\square$

## C.6 IDENTIFIABILITY, CALIBRATION, AND ROBUSTNESS

**Identifiability.** The decomposition into $\rho_d$ and $d_r$ is not unique across encoders and pooling rules. Fixing $L^*$, pooling, whitening, and per-dev $z$-scoring yields conditionally comparable scalars that support within-paper tests, not universal constants.

**Calibration.** Logistic parameters $(\alpha, \beta)$ depend on model, layer, and prompt format. We fit them on a dev subset and report generalization on held-out tasks.

**Robustness.** Appendix F specifies ablations over layer readout, pooling, metric choice, and a frozen external encoder. Signs and orderings are expected to be stable; magnitudes can vary.

## C.7 LIMITATIONS AND SCOPE

- The logistic surrogate is phenomenological: it summarizes sharpness and midpoints; it is not a mechanistic derivation.

- $S$ is a predictive correlate calibrated by whitening and $z$-scoring; it is not an absolute energy.

- Results are scoped to the tasks, models, and prompts studied; extension to tool use, multi-step reasoning, and long-context settings is future work.

| Symbol | Meaning / where used |
|---|---|
| $\rho_d$ | Within-target pattern density (cohesion) on whitened span embeddings; standardized via per-dev $z$-scoring. Used in $S$ and E2/E3. |
| $d_r$ | Prior–target representational mismatch (e.g., $1 - \cos(\mathbf{e}_{\text{prior}}, \mathbf{e}_T)$ or an $L^2$ variant); standardized via per-dev $z$-scoring. |
| $k$ | Number of few-shot exemplars (anchors) in the prompt; budget term in $S$. |
| $S$ | Anchoring score: $S = \tilde{\rho}_d - \tilde{d}_r - \eta \log k_{\text{eff}}$ (few-shot: $k_{\text{eff}}{=}k$; zero-shot: $k_{\text{eff}}{=}1$; default $\eta{=}1$). Used as a predictive correlate. |
| $S_c$ | Task/model-dependent critical region for threshold-like behavior; estimated via logistic calibration. |
| $k_{50}$ | E2 midpoint in *shots*: the $k$ where success reaches $50\%$ under a logistic fit (E2 only). |
| $\theta_{50}$ | E3 internal midpoint in *shots*, estimated from E3's own sweep; distinct from $k_{50}$. |
| $\widehat{S}_{\max}$ | Peak layer-wise anchoring: $\max_\ell S^{(\ell)}$ (after whitening and per-dev $z$). |
| $\text{AUS}_N$ | Normalized area under $S^{(\ell)}$ across layers: $\text{AUS}_N = \frac{1}{L} \sum_{\ell=1}^{L} S^{(\ell)}$. |

*Calibration:* whitening and per-dev $z$-scaling are defined in Appendix D.

# D  CONFIGURATION AND PREPROCESSING

**Development (dev) pool.**  For each backbone, build a fixed dev pool of span embeddings used only for calibration. Construct 1,000 prompts with $k{=}3$ examples (mixed tasks or the E3 behavioral task), using the same layer $L^*$ and pooling as used in geometry. The dev pool must not overlap the test items. Release indices and seeds.

**Whitening of span embeddings.**  Let $\{\mathbf{e}_j\}_{j=1}^{N}$ be span embeddings (instruction and example spans) from the dev pool at layer $L^*$, mean-pooled and unit-normalized. Compute the dev mean and a shrunk covariance

$$\boldsymbol{\mu} = \tfrac{1}{N} \sum_j \mathbf{e}_j, \qquad \boldsymbol{\Sigma} = \tfrac{1}{N} \sum_j (\mathbf{e}_j - \boldsymbol{\mu})(\mathbf{e}_j - \boldsymbol{\mu})^\top + \lambda \mathbf{I},$$

with $\lambda{=}10^{-5}$ by default. Form the whitening matrix $\mathbf{W} = \boldsymbol{\Sigma}^{-1/2}$ using eigendecomposition or Cholesky on $\boldsymbol{\Sigma}$; clip eigenvalues below $\lambda$ for numerical stability before inversion. For any span embedding $\mathbf{e}$, use the whitened vector

$$\mathbf{e}^{(\text{whitened})} = \mathbf{W}\,(\mathbf{e} - \boldsymbol{\mu}).$$

All geometry computations (pairwise distances and centroids) use whitened embeddings.

**Per-dev $z$-scaling of scalar metrics.**  For each layer $\ell$, compute on the dev pool the moments

$$\mu_{\rho_d}^{(\ell)}, \ \sigma_{\rho_d}^{(\ell)} \quad \text{for } \rho_d^{(\ell)}, \qquad \mu_{d_r}^{(\ell)}, \ \sigma_{d_r}^{(\ell)} \quad \text{for } d_r^{(\ell)}.$$

Standardize any run's measurements by

$$\tilde{\rho}_d^{(\ell)} = \frac{\rho_d^{(\ell)} - \mu_{\rho_d}^{(\ell)}}{\sigma_{\rho_d}^{(\ell)} + \epsilon}, \qquad \tilde{d}_r^{(\ell)} = \frac{d_r^{(\ell)} - \mu_{d_r}^{(\ell)}}{\sigma_{d_r}^{(\ell)} + \epsilon},$$

with $\epsilon{=}10^{-8}$. Optionally clip standardized values to $[-5, 5]$.

**Anchoring score (used in E3).**  With whitened embeddings and per-dev $z$-scores,

$$S^{(\ell)} = \tilde{\rho}_d^{(\ell)} - \tilde{d}_r^{(\ell)} - \log k_{\text{eff}}, \qquad k_{\text{eff}} = \begin{cases} k & \text{few-shot,} \\ 1 & \text{zero-shot.} \end{cases}$$

In E3 we fix $k_{\text{eff}}$. If $k$ varies across runs, standardize $\log k$ on the dev pool as well. Summaries are

$$\widehat{S}_{\max} = \max_\ell S^{(\ell)}, \qquad \text{AUS}_N = \tfrac{1}{L} \sum_{\ell=1}^{L} S^{(\ell)},$$

where $L$ is the number of layers.

**Distance and density proxies.** Given a set of $k$ anchor spans with whitened embeddings $\{\mathbf{e}_i\}_{i=1}^k$ and centroid $\mathbf{e}_T = \frac{1}{k}\sum_i \mathbf{e}_i$,

$$\rho_d(P_T) = \left[\tfrac{1}{\binom{k}{2}}\sum_{i<j}\|\mathbf{e}_i - \mathbf{e}_j\|_2\right]^{-1}, \qquad d_r(P_{\text{prior}}, P_T) = 1 - \cos(\mathbf{e}_{\text{prior}}, \mathbf{e}_T),$$

where $\mathbf{e}_{\text{prior}}$ is the mean of $m$ zero-shot encodings with fixed seeds. We also report an $L^2$ variant $d_r^{(\text{L2})} = \|\mathbf{e}_{\text{prior}} - \mathbf{e}_T\|_2$ for completeness.

**Pooling and layer selection.** Use the same $L^*$ and pooling for dev and test. Unless specified otherwise, pooling is mean over span tokens at layer $L^*$. Sensitivity to $L^*$ and pooling is reported in the main text and the geometry appendix.

**Backbones and seeds.** For API-based evaluations use fixed model versions per backbone and record the provider version string, temperature, and seed. For local evaluations record the checkpoint hash, tokenizer version, and decode parameters. Log all random seeds that control prompt sampling, data synthesis, and evaluation order.

**Reproducibility checklist.** Release: (i) dev indices and seeds, (ii) $(\boldsymbol{\mu}, \boldsymbol{\Sigma})$ statistics per backbone, (iii) code to compute whitening and per-dev $z$-scores, (iv) the exact prompt templates and anchoring formats, and (v) the list of model versions and decoding parameters.

# E AMBIGUOUS ANCHORS AND PATTERN SELECTION

## E.1 AMBIGUOUS PATTERN TEST RESULTS

When presented with ambiguous anchors in Case Study #1, different models favor different latent pattern interpretations, consistent with threshold-crossing behavior near the activation boundary.

**Test setup.**

Example 1: 33 - 27 = 60     Example 2: 11 - 9 = 20
**Question:** 57 - 81 = ?

Table 4: Model responses under ambiguous anchors and inferred pattern hypotheses.

| Model | Answer | Pattern | Rule |
|-------|--------|---------|------|
| M1 | 240 | $P_{\text{abs-mult}}$ | $|a - b| \times 10$ |
| M2 | 138 | $P_{\text{add}}$ | $a + b$ |
| M3 | 138 | $P_{\text{add}}$ | $a + b$ |
| M4 | $-240$ | $P_{\text{signed-mult}}$ | $(a - b) \times 10$ |

## E.2 ANCHORING ANALYSIS

With identical $k = 2$ but ambiguous exemplars, multiple target hypotheses $P_T^{(h)}$ are plausible because the representational gap $d_r(P_{\text{prior}}, P_T^{(h)})$ differs across interpretations. Using the anchoring score,

$$S = \rho_d(P_T) - d_r(P_{\text{prior}}, P_T) - \log k,$$

and $k$ fixed, the $-\log k$ term is constant across models. Differences in $\rho_d$ and $d_r$ determine which hypothesis crosses the threshold.

**Hypothesis $P_{\text{abs-mult}}$:** $|a - b| \times 10$ **(M1).**

- Small $d_r$: both anchors map exactly, $|33 - 27| \to 60$, $|11 - 9| \to 20$.
- Moderate $\rho_d$: absolute value and multiplication features are well represented.
- Outcome: $S$ exceeds $S_c$; prediction $|57 - 81| \times 10 = 240$.

**Hypothesis $P_{\mathbf{add}}$: $a + b$ (M2, M3).**

- Small $d_r$: both anchors map exactly, $33 + 27 = 60$, $11 + 9 = 20$.
- High $\rho_d$: addition is a dominant arithmetic prior.
- Outcome: high density favors this hypothesis; prediction $57 + 81 = 138$.

**Hypothesis $P_{\mathbf{signed\text{-}mult}}$: $(a - b) \times 10$ (M4).**

- Fits anchors: $(33 - 27) \times 10 = 60$, $(11 - 9) \times 10 = 20$.
- Moderate $\rho_d$ with a larger effective $d_r$ than $P_{\mathrm{add}}$.
- Outcome: crosses threshold with sign preserved; prediction $(57 - 81) \times 10 = -240$.

### E.3 THRESHOLD-CROSSING TAKEAWAY

With $k = 2$, the success probability is near the critical region. Small model-specific differences in $\rho_d$ and $d_r$ tilt the decision among competing hypotheses, consistent with

$$S(\mathcal{A}) \approx S_c \quad \Longrightarrow \quad \text{qualitatively different bindings across models.}$$

This supports the threshold-crossing view: near the boundary, marginal variations in the latent repository produce divergent semantic interpretations under the same ambiguous anchors.

## F EXPERIMENT PART 3: GEOMETRIC TRAJECTORY ANALYSIS (WILL REVISE FOR CAMERA-READY)

### GOALS

**E3 tests a geometric implication of** UCCT**.** The aims are to:

1. Measure layer-wise *anchoring geometry* via $S^{(\ell)}$ along the prompt trajectory and summarize runs by peak anchoring $\widehat{S}_{\max}$ and normalized area $\mathrm{AUS}_N$.

2. Relate geometry to behavior by estimating an *internal few-shot midpoint* $\theta_{50}$ in a self-contained sweep, and assess whether geometry *predicts* midpoints (as a correlate, not a causal effect).

### SETUP

**Backbones.** Three instruction-tuned decoder-only LLMs of similar scale (e.g., Meta-LLaMA-3.1-8B, Phi-4, Gemma-3-4B-it).[3]

**Tasks and prompts.** For qualitative overlays, we build 25 evaluation prompts spanning commonsense (Talmor et al., 2019), logical inference (Liu et al., 2020), science/knowledge (Clark et al., 2018), arithmetic, and code (Huang et al., 2025). For geometry→behavior correlates, we use a *single* behavioral testbed: two-digit *decimal* addition with an explicit tag [task=add] to avoid token ambiguity.

**Replication.** Unless noted, each (model, configuration) is evaluated over 10 seeds with re-sampled few-shot sets.

### METRICS

$$\rho_d^{(\ell)} = \left[ \tfrac{1}{\binom{k}{2}} \sum_{i<j} \|\mathbf{e}_i^{(\ell)} - \mathbf{e}_j^{(\ell)}\|_2 \right]^{-1} \quad \text{(cohesion)},$$

$$d_r^{(\ell)} = 1 - \cos\!\big(\mathbf{e}_{\mathrm{instr}}^{(\ell)}, \mathbf{e}_T^{(\ell)}\big), \quad \mathbf{e}_T^{(\ell)} = \tfrac{1}{k} \sum_i \mathbf{e}_i^{(\ell)} \quad \text{(mismatch)},$$

$$S^{(\ell)} = \tilde{\rho}_d^{(\ell)} - \tilde{d}_r^{(\ell)} - \log k_{\mathrm{eff}}, \qquad k_{\mathrm{eff}} = k \text{ (few-shot) or } 1 \text{ (zero-shot)}.$$

---

[3]If a backbone is unstable, substitute a similarly sized instruction-tuned model and record the change in the artifact bundle.

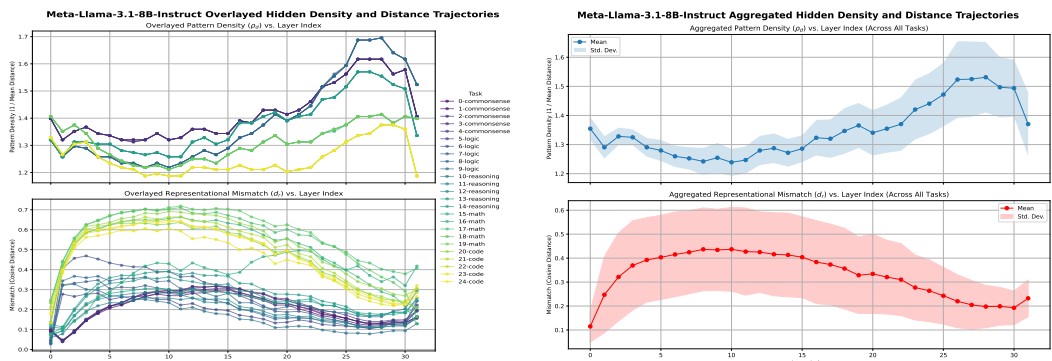

Figure 4: **Meta-LLaMA-3.1-8B.** Layer-wise cohesion $\rho_d^{(\ell)}$ and mismatch $d_r^{(\ell)}$ across representative tasks (mean $\pm 1$ sd).

Tildes denote per-dev $z$-scores (Appendix D). We also report $d_r^{(\ell,\text{L2})} = \|\mathbf{e}_{\text{instr}}^{(\ell)} - \mathbf{e}_T^{(\ell)}\|_2$ for robustness.

**Anchoring strength (E3 notation).**   We use a budget-weighted variant

$$S \;=\; \tilde{\rho}_d \;-\; \tilde{d}_r \;-\; \eta \log k_{\text{eff}}, \qquad k_{\text{eff}} = \begin{cases} k & \text{few-shot,} \\ 1 & \text{zero-shot.} \end{cases}$$

We set $\eta{=}1$ throughout this paper, so the expression reduces to the form used in the main text; we introduce $\eta$ only to make the budget term explicit for future work.

PROTOCOL

**Overlay protocol (qualitative geometry).**   For each evaluation prompt we prepend three domain-matched demonstrations and record $\rho_d^{(\ell)}$ and $d_r^{(\ell)}$ across layers for all backbones; we show both per-model overlays and aggregates.

**Correlate protocol (geometry→behavior).**   For each (model, seed) in the addition testbed, we construct a fixed $k{=}3$ few-shot prompt *without labels*, compute per-layer $\rho_d^{(\ell)}$ and $d_r^{(\ell)}$ over instruction and example spans, and form $S^{(\ell)}$ using whitening and dev-set $z$-scaling. We summarize geometry by $\widehat{S}_{\text{max}} = \max_\ell S^{(\ell)}$ and $\text{AUS}_N = \frac{1}{L}\sum_\ell S^{(\ell)}$. Independently, we fit a logistic to accuracy-vs.-shots within this E3 testbed to obtain $\theta_{50}$ (Eq. (3)).

RESULTS

**Three-stage layer trajectory.**   Across backbones and tasks we observe:

- *Early enrichment:* $\rho_d^{(\ell)}$ dips as examples specialize.

- *Mid-layer alignment:* $d_r^{(\ell)}$ falls while $\rho_d^{(\ell)}$ recovers.

- *Late standardization:* formatting/serialization re-clusters spans near output.

**Minimal quantitative readout.**   At the run level (model by seed), larger $\widehat{S}_{\text{max}}$ is associated with smaller $\theta_{50}$, while $\text{AUS}_N$ shows a weaker association. Signs are consistent across backbones. Robustness overlays appear in Figs. 4–6.

NEGATIVE CONTROLS AND ROBUSTNESS

**Negative controls.**   Dense but off-task anchors yield high cohesion *and* high mismatch; behavior does not improve, consistent with mismatch dominating $S$. At approximately matched mismatch, increasing cohesion lowers the internal midpoint, consistent with density's role in $S$.

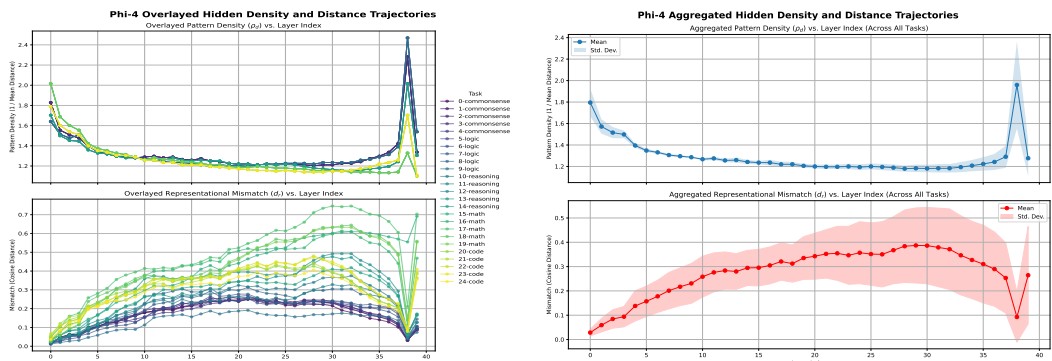

Figure 5: **Phi-4.** Same readout as Fig. 4. U-shaped cohesion with falling mismatch; peak depth varies by model.

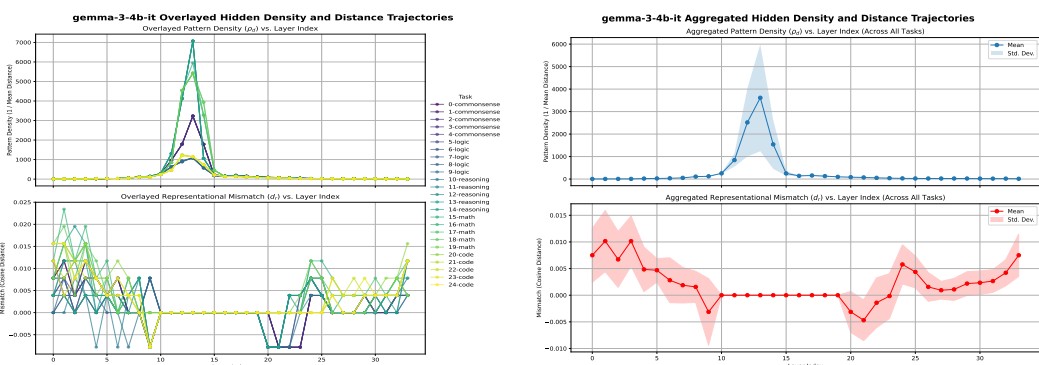

Figure 6: **Gemma-3-4B-it.** Same readout. Despite scale differences in $d_r$ and $\rho_d$, the three-stage pattern and the $S^{(\ell)}$ peak remain visible.

**Robustness.** The negative association between $\widehat{S}_{\max}$ and $\theta_{50}$ holds under last-token vs. mean pooling, $L^2$ vs. cosine, and with a frozen external encoder. Calibration/whitening follow Appendix D.

LIMITATIONS

- **Correlational.** Geometry-to-behavior relations are predictive correlates, not causal proofs.

- **Span/metric choices.** We report mean pooling and cosine by default; ablations mitigate but do not exhaust alternatives.

- **Coverage.** Three backbones and 25 tasks; scaling to more families and sizes is future work.

FUTURE WORK (E3-SPECIFIC)

- Quantify the correlate on at least one backbone: regress $\theta_{50}$ on $\widehat{S}_{\max}$ and report slope with bootstrap CI, $R^2$, and Spearman $\rho$.

- Minimal robustness toggle (e.g., last-token vs. mean pooling or cosine vs. $L^2$) and confirm slope sign stability.

- Negative-control stress test: matched mismatch with varied cohesion and report $\Delta\theta_{50}$ with CI.

- Export seed-wise artifacts (`thresholds.csv`, `geometry.csv`, `geometry_summaries.csv`) and a `metadata.json` with model versions and seeds.

POINTER FOR THE MAIN TEXT

*Appendix F (E3) shows that peak layer-wise anchoring $S^{(\ell)}$ correlates with internal few-shot midpoints $\theta_{50}$, providing a geometry-to-behavior correlate consistent with UCCT.*

# G   Vision Example

Figure 7 illustrates that UCCT's threshold-crossing principles apply across modalities. In a 4-shot setup, unlabeled pretraining provides distributed visual features ($P_{\text{prior}}$), the labeled *cat* shots establish within-class cohesion $\rho_d(P_T)$, and anchoring strength $S(\mathcal{A})$ exceeds a task-dependent critical value $S_c$, yielding reliable classification. This example is illustrative and complements the quantitative results in the main text.

**Scope note.**   We use human learning as an analogy for *selective access to latent structure*, not as a claim of neural equivalence.

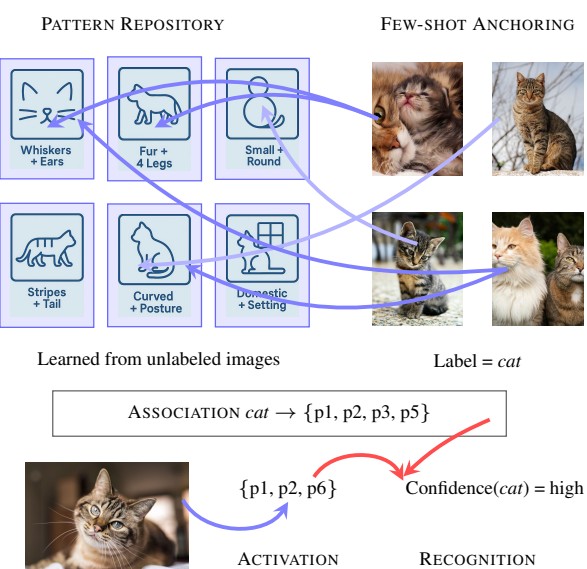

Figure 7: **Cross-modal anchoring:** Few-shot examples bind visual patterns to a semantic label.

**Developmental parallel as analogy.**   A common objection is that a young child can recognize cats after a small number of labeled examples, while modern models require large-scale pretraining. In our lens, this contrast reflects differences in how much *unlabeled* experience accumulates before semantic binding, not a fundamentally different mechanism of anchoring.

**Unlabeled accumulation precedes rapid labeling.** Before labels are introduced, systems can acquire extensive latent structure from unlabeled exposure. In humans, early experience provides rich perceptual regularities; in models, pretraining plays a similar role by building $P_{\text{prior}}$. When labels arrive, a few coherent examples can bind existing patterns to a semantic target, consistent with threshold-like behavior.

**A small anchor can flip behavior when $\rho_d$ is high.** If preexisting visual features for cat-like regularities are dense (high $\rho_d$) and close to the intended target (small $d_r$), then only a small number of labeled shots is needed. In UCCT terms, $S = \rho_d - d_r - \log k$ can exceed $S_c$ with modest $k$, producing a rapid rise in accuracy.

**Models exhibit the same qualitative signature.** In the 4-shot vision example, the shots act as anchors that increase effective target density and reduce prior–target mismatch. The transition from unreliable to reliable classification follows the same signature as in E1 and E2, now in a different modality. Full image sets, prompts, and per-seed outcomes are provided with the code release.