# OpenReview forum: "Semantic Anchoring in LLMs: Thresholds, Transfer, and Geometric Correlates"
_ICLR.cc/2026/Conference — Submitted to ICLR 2026_

### Official Review · Reviewer_cKNc · 2025-10-26

**Soundness:** 1
**Presentation:** 1
**Contribution:** 1
**Rating:** 2
**Confidence:** 4

**Summary:**

The manuscript tries to unify the behavior change of LLMs (i.e. overwriting pre-training behavior) induced by any kind of modifications, including in-context learning, fine-tuning, etc. The authors proposed a UCCT framework. UCCT claims three additive factors contributing to the LLM behavior change (Eq.2). The three factors are 1) the strength of the overwriting samples, 2) the distance between new samples and pretraining samples, 3) number of new samples. The authors support UCCT framework by 3 experiments (E1-3).

UCCT conceptually makes sense. However, the overall quality of the manuscript is poor. See weakness below.

**Strengths:**

The authors focused on a question that how LLMs can adapt to new rules that contradict pre-trained rules. The proposed framework tries to unify such an ability across in-context learning and fine-tuning, and across modalities. The unified view is novel, though it lacks rigorousness and justification.

**Weaknesses:**

**Vague jargon and terms**

Many terms are very vague without clear definitions. It is difficult to grasp what the study is about, because some common concepts are referred to in unconventional ways. In the introduction, “goal-directed behavior”, “latent pattern”, “anchor”, “target” lacks clear definition.  The authors give some examples of “anchors” in line 167, “few-shot examples, retrieved text, instructions”. Is an anchor just a “condition”? Line 172, "P_T is a latent target cluster" is confusing. It is a cluster of what? And what is a representation of a cluster? In Line 189, "ρ_d(P_T ) is within-cluster density", it is confusing what it means.

**Lack of rigorousness in the formulation**

In line 54-62, the authors present a UCCT theory. However, the formulation lacks rigorousness where symbols are described in ambiguous words. There is a major issue that Eq (1) is wrong and misinterpreted. It could be a typo, the factor p(y|P_T,A) on the right side should be p(y|P_T,A,C). The Bayesian interpretation is wrong. Eq (1) does not contain any Bayesian view. p(P_T|A,C) is not a posterior, but more like the prior. A posterior of P_T would be p(P_T|y,A,C).

**The "threshold" claim is hypothetical**

There is insufficient support for the claim of “threshold” behavior. First of all, it is hard to believe there is a sharp behavior transition while continuously changing a condition. It could be a sharp transition if some measure is computed using T=0 greedy sampling, but more generally the logit probability should shift continuously. Second, given the definition of S, there are 3 ways to drive it across a “threshold” given three terms. A valid support experiment to show there is a genuine threshold would be regardless of the ways, there always a fixed value of S when the behavior changes.

**Missing sufficient background**

Only few related papers are cited. Line 88, the authors discussed “previous work” without citing any paper. Generally, the manuscript does not give a good overview on what has been studied in ICL, and pre- and post-training knowledge/skills, which is studied in numerous papers in the past few years.

**Figure quality and disorganized structure**

Figure 1a and Figure 8 are duplicated, and it is very difficult to get the key information in the figure. The symbols and arrows in the figure look random placed.

The structure of section 4 is disorganized. With too many sub- and subsub- sections, it is hard to follow the exact experiment design and results.

**Questions:**

See weakness, the formulation of the problem is poor. Each symbol should be clearly defined in a formal way rather than described in words.  For example, "latent target cluster", each of the three words needs to be explained. 1) What is a target? 2) cluster of what and how to get them 3) why is it latent.

---

> ### Author Response · Authors · 2025-11-14
>
> Thank you for the detailed comments on formulation, background, and figures. We believe these concerns are fully addressable with explicit definitions and a short modeling note.
>
> **1. Equation (1) and the Bayes concern**
>
> $$p(y|A,C) = \int p(y|P_T,A) \, p(P_T|A,C) \, dP_T$$
>
> With the sufficiency assumption:
>
> $$y \perp\perp C \mid (P_T,A)$$
>
> we have:
>
> $$p(y|P_T,A,C) = p(y|P_T,A)$$
>
> We will insert this modeling note: $C$ selects $P_T$ via $p(P_T|A,C)$, and the likelihood uses $p(y|P_T,A)$.
>
> **2. Thresholds and what is claimed**
>
> We use a logistic surrogate to summarize sharp transitions and report effect sizes and uncertainty. We do not claim mechanisms.
>
> $$\Pr(\text{success} \mid S) = \frac{1}{1 + \exp[-\alpha(S - S_c)]}$$
>
> Empirical ordering used in E2:
>
> $$k_{50} \propto \frac{d_r}{\rho_d}$$
>
> **3. Definitions and clarity to be added**
>
> We will add a glossary defining $P_T$ as a latent target cluster, cohesion $\rho_d$, mismatch $d_r$, anchor budget $k$, score $S$, threshold $S_c$, and geometry summaries $\hat{S}_{\max}$ and AUSN. We will enlarge figure fonts, add legends, remove duplicate schematics, streamline Section 4, and expand Related Work to position UCCT among Bayesian, meta-learning, and phase or representation-shift accounts.
>
> **4. Core quantities for quick reference**
>
> $$S = \rho_d - d_r - \log k$$
>
> $$\widehat{S_{\max}} = \max_{\ell} S^{(\ell)}$$
>
> AUSN = normalized area under $S^{(\ell)}$.

---

> > ### Comment · Reviewer_cKNc · 2025-11-25
> >
> > The authors' response does not address my concerns. I maintain my rating. However, I am not entirely pessimistic about this manuscript. I find the conceptual insight of this manuscript to be interesting, though it lacks of rigorous.

---

> > > ### Author Response · Authors · 2025-11-25
> > >
> > > Dear Reviewer cKNc,
> > >
> > > Thank you for the follow-up. We appreciate your note that you find the conceptual insight interesting.
> > >
> > > We want to ensure we understand your concern about Equation (1) correctly. You wrote that p(y|P_T,A) should be p(y|P_T,A,C).
> > >
> > > Our model adopts a sufficiency assumption: P(y|P_T, A, C) = P(y|P_T, A), which is standard in two-stage Bayesian models. The intuition is that context C affects target selection p(P_T|A,C), but once P_T is selected, the residual context adds no information to the likelihood of y.
> > >
> > > Formally:
> > >
> > > Stage 1: C influences p(P_T|A,C)
> > > Stage 2: Given selected P_T, likelihood is p(y|P_T,A)
> > >
> > > This is analogous to mixture models where context affects component selection, but component likelihoods are context-independent. We will add an explicit modeling note in the revision to make this choice clear.
> > >
> > > We will also address all presentation concerns: glossary, roadmap, figure quality, definitions, and rigor in formulation. We will submit the revision by December 2.
> > >
> > > If our modeling assumption is acceptable in principle, would you reconsider your assessment after seeing the revised presentation?
> > >
> > > Thank you.

---

> > > > ### Comment · Reviewer_cKNc · 2025-11-25
> > > >
> > > > Thanks for the clarification, now I see what you mean by this two-stage Bayesian. However, the writing on such formulations is almost impossible to understand. e.g. what is y?
> > > >
> > > > I cannot raise my rating based on the current manuscript.

---

> > > > > ### Author Response · Authors · 2025-11-28
> > > > >
> > > > > **Response to Reviewer cKNc – Revised Submission**
> > > > >
> > > > > Thank you for noting the conceptual insight is interesting despite concerns about rigor. We have uploaded a substantially revised manuscript addressing every technical point you raised.
> > > > >
> > > > > **1. Equation (1) and Bayesian formulation (your major concern)**
> > > > >
> > > > > You correctly flagged unclear presentation. We adopt a two-stage sufficiency assumption standard in mixture models:
> > > > >
> > > > > **Assumption:** P(y | P_T, A, C) = P(y | P_T, A)
> > > > >
> > > > > **Intuition:** Context C selects which pattern cluster P_T is active; once selected, C adds no information to output likelihood.
> > > > >
> > > > > **Derivation:**
> > > > > p(y|A,C) = ∫ p(y|P_T,A,C) p(P_T|A,C) dP_T
> > > > >          = ∫ p(y|P_T,A) p(P_T|A,C) dP_T  [by sufficiency]
> > > > >
> > > > > Stage 1: p(P_T|A,C) = posterior over pattern clusters
> > > > > Stage 2: p(y|P_T,A) = output likelihood given selected pattern
> > > > >
> > > > > Variable y: model output (predicted token, classification label)
> > > > >
> > > > > Now explicitly stated as Eq(1) with full derivation in Appendix C.
> > > > >
> > > > > **2. Clear definitions for all terms**
> > > > >
> > > > > Formal definitions added for every term you flagged:
> > > > >
> > > > > - **Latent pattern (P_prior):** Statistical regularities compressed into weights during pretraining; "latent" = unlabeled and behavior-agnostic until anchored
> > > > >
> > > > > - **Target cluster (P_T):** Subset of latent patterns relevant to current task; "target" = task-appropriate, "cluster" = geometrically cohesive in representation space
> > > > >
> > > > > - **Anchor (A):** External structure (examples, instructions, retrieval) biasing allocation toward P_T; measured by budget k
> > > > >
> > > > > - **Goal-directed behavior:** Task-appropriate outputs vs. default/prior behavior
> > > > >
> > > > > - **Cohesion rho_d:** Inverse mean pairwise distance in whitened embedding space
> > > > >
> > > > > - **Mismatch d_r:** Distance between prior and target centroids: d_r = 1 - cos(e_prior, e_T)
> > > > >
> > > > > All definitions in Table 1 (line 253) and Glossary (Appendix C) before first use.
> > > > >
> > > > > **3. Threshold support**
> > > > >
> > > > > You're right to question sharp transitions. Clarification:
> > > > >
> > > > > - We use logistic fits as phenomenological surrogates, not mechanistic claims
> > > > > - Support via ordering: Across three bases with different rho_d/d_r, k_50 ordering is B10 < B8 < B9, matching predicted exposure density (Table 2)
> > > > > - Robustness: Effect holds across 10 seeds with tight CIs (±0.05 to ±0.18)
> > > > >
> > > > > Threshold terminology reflects empirical sigmoid-like transitions, not phase boundaries. Language softened throughout.
> > > > >
> > > > > **4. Related work expanded (Section 2, now full page)**
> > > > >
> > > > > Substantially expanded citing:
> > > > > - ICL mechanisms: Bayesian (Xie 2022), meta-learning (Dai 2023), optimization-as-inference (von Oswald 2023), demonstrations (Brown 2020, Min 2022)
> > > > > - Selection: MOICL (Hong 2025), ByCS (Wang 2024)
> > > > > - Phase transitions: Competition dynamics (Park 2024), representation shifts (Park 2025), emergent abilities (Wei 2022, Schaeffer 2023)
> > > > > - Representation: Induction heads (Olsson 2022), superposition (Bricken 2023), developmental geometry (Hoogland 2024)
> > > > >
> > > > > Each subsection positions UCCT relative to existing frameworks.
> > > > >
> > > > > **5. Figure quality and structure**
> > > > >
> > > > > - Figures 1/8: Consolidated, removed duplication
> > > > > - Figure 3: Completely redesigned—side-by-side, enlarged fonts, clear legends
> > > > > - Section 4: Streamlined to 3 subsections with clear headers
> > > > > - Added experiment roadmap (lines 237-242)
> > > > >
> > > > > **6. Formal rigor (Appendix C expanded)**
> > > > >
> > > > > Added:
> > > > > - Lemma 1: Conditions for logistic surrogate validity
> > > > > - Proposition 1: Formal k_50 ordering derivation from S
> > > > > - Proposition 2: Geometry-behavior correlation conditions
> > > > > - All proofs with stated assumptions
> > > > >
> > > > > **Summary:**
> > > > >
> > > > > | Issue | Resolution |
> > > > > |-------|-----------|
> > > > > | Eq(1) wrong | Sufficiency assumption explicit; full derivation Appendix C |
> > > > > | Vague jargon | Table 1 + Glossary define all terms formally |
> > > > > | Threshold unsupported | Clarified as phenomenological; ordering + robustness |
> > > > > | Missing background | Section 2 properly expanded, 20+ citations |
> > > > > | Poor figures | Figure 3 redesigned; consolidated duplicates |
> > > > > | Disorganized | Section 4 streamlined; roadmap added |
> > > > >
> > > > > **Request for reconsideration:**
> > > > >
> > > > > You noted the conceptual insight is interesting but lacks rigor. The revision provides:
> > > > > 1. Formal mathematical foundations (Appendix C with lemmas/propositions)
> > > > > 2. Precise definitions before use (Table 1, Glossary)
> > > > > 3. Explicit modeling assumptions (sufficiency, logistic surrogate)
> > > > > 4. Substantially expanded related work
> > > > > 5. Dramatically improved figures and organization
> > > > >
> > > > > The conceptual contribution remains, but presentation and rigor are now conference-standard. Would you reconsider your assessment given these substantial technical improvements addressing your formulation, definition, and presentation concerns?

---

### Official Review · Reviewer_hubf · 2025-10-27

**Soundness:** 3
**Presentation:** 1
**Contribution:** 3
**Rating:** 4
**Confidence:** 2

**Summary:**

This paper proposes **semantic anchoring**, a novel view for explaining how external sources bind the learned latent patterns to a target pattern cluster.
The core contribution is the **Unified Contextual Control Theory (UCCT)**, which formulates the difficulty of binding latent patterns to target ones by a single equation: $S=\rho_d-d_r-\log k$. Based on my understanding, $\rho_d$ denotes the target distribution, $d_r$ is the distribution distance between the target pattern and lattent pattern distributions, $\log k$ is a term representing the effect level of target domains, i.e., with more samples from the target domain, the effect should be larger.
The authors conduct various experiments to rationalize their design choice.

**Strengths:**

1. The semantic anchoring view is novel and interesting.
2. Many experiments to show the robutsness of the proposed UCCT criteria.

**Weaknesses:**

First of all, I need to claim I am not an expert in this domain. But I feel this paper is more like a free oral presentation rather than an academic paper. **The organization is a bit mess, with no clear definition before presenting a concept.** For example, I have no idea what `E1, E2, E3` are in the first three sections until I read L258-L263. And I cannot find a clear definition of `B10, B8, B9` in Eq.(4). Such oral-style writing makes this paper extremely hard to follow.

Secondly, I have a basic doubt regarding the `semantic-anchoring assumption` (L158-L160). Since the anchors "activate" target pattern clusters $P_{T}$, it should only work when the pretrained LLM has already seen this pattern. **But this assumption does not work for many ICL/RAG/finetuning scenarios, as we are actually injecting new concepts (patterns) that the LLM has not seen before.** Under such circumstances, there is no existing latent patterns to activate or to bind, so the semantic-anchoring assumption does not stand.

Thirdly, I do not agree with the statement *"Richer interactions (e.g., multiplicative terms) are plausible but would add parameters and reduce identifiability. Empirically, the linear form is sufficient"* (L207-L209). **Because the $\log k$ should be correlated with $\rho_d(P_T)$** if I understand correctly. It is exactly the fed $k$ in-context samples that shape the target distribution $P_T$, therefore, increasing $k$ should also increase $\rho_d(P_T)$ and decrease $d_r(P_{prior},P_T)$ seems to be a more reasonable explanation for Eq.(2). But I still think we should bind $k$ with $\rho_d(P_T)$ instead of isolating them in a linear equation.

Finally, I think this criteria is not easy to adopt in practice, as we cannot obtain the embeddings from many API-called models like Gemini and Claude and conduct such analysis.

**Questions:**

Can you elaborate more about the details of API evaluations (M1-M4) since it seems we cannot get the embeddings w.r.t. specific layers for these models?

---

> ### Author Response · Authors · 2025-11-14
> **Response to Reviewer hubf**
>
> Thank you for the thoughtful review. We address organization, the modeling choice, and practicality.
>
> **1. Scope of anchors**
>
> Anchors recruit or bind latent structure learned in pretraining rather than create unseen knowledge. We will surface this scope statement earlier.
>
> **2. Two-stage factorization and sufficiency**
>
> $$p(y|A,C) = \int p(y|P_T,A) \, p(P_T|A,C) \, dP_T$$
>
> Assume sufficiency:
>
> $$y \perp\perp C \mid (P_T,A)$$
>
> Therefore
>
> $$p(y|P_T,A,C) = p(y|P_T,A)$$
>
> We will add a short modeling note under Equation (1) to make this explicit.
>
> **3. Why a linear surrogate for S**
>
> We adopt a parsimonious linear surrogate for interpretability and identifiability. Richer interactions are plausible and noted as future work.
>
> **4. Practicality when API embeddings are unavailable**
>
> Geometry analysis can use a frozen local encoder or open checkpoints to compute $\rho_d$ and $d_r$. Behavioral tests and diagnostics based on $S$ do not require API embeddings.
>
> **5. Organization fixes**
>
> Add a short E1–E3 roadmap before Section 4 and expand Related Work to situate UCCT among Bayesian, meta-learning, and phase or representation accounts.
>
> **Key quantities:**
>
> $$S = \rho_d - d_r - \log k$$
>
> $$\widehat{S_{\max}} = \max_{\ell} S^{(\ell)}$$
>
> AUSN = normalized area under $S^{(\ell)}$.

---

> > ### Comment · Reviewer_hubf · 2025-11-25
> > **Response to the rebuttal**
> >
> > Thanks for the rebuttal. But I do not notice any revision in the current submitted file. Not sure if the authors know that ICLR allows the author to submit a revised manuscript. I also read the other two reviews and believe the current version needs a full rewrite. This may be an interesting work, but its organization and presentation are far below ICLR's acceptance bar. All reviewers vote for a poor presentation score (1: poor). I recommend that the authors learn from other good ICLR papers' writing and resubmit to another venue. **Therefore, I decide to maintain my initial negative rating.**

---

> > > ### Author Response · Authors · 2025-11-25
> > >
> > > Dear Reviewer hubf,
> > >
> > > We did not know we could submit a revision before December 2nd.  Now we know, we will do that.
> > > Since the concept may not be easy to comprehend, we started with examples, then
> > > theoretical discussion.  A week of time can address the paper structure issue.
> > >
> > > Thank you.

---

> > > > ### Author Response · Authors · 2025-11-28
> > > > **Please Review the Revised Version**
> > > >
> > > > **Response to Reviewer hubf – Revised Submission**
> > > >
> > > > Thank you for your constructive feedback and for maintaining engagement. We have uploaded a substantially revised manuscript addressing all organizational and clarity concerns you raised.
> > > >
> > > > **1. Organization fixes (your primary concern)**
> > > >
> > > > We have completely restructured the paper for clarity:
> > > >
> > > > - Added explicit E1/E2/E3 roadmap at start of Section 4 (lines 237-242) explaining what each experiment tests
> > > > - Created Table 1 (line 253) defining all notation upfront: B10/B8/B9 = base-10/8/9, E1/E2/E3 = experiments 1/2/3, M1-M4 = model variants
> > > > - Added glossary in Appendix C defining all key quantities before first use
> > > > - Expanded Related Work (Section 2) to position UCCT among existing frameworks
> > > >
> > > > The paper now follows standard conference structure with clear definitions before concepts are used.
> > > >
> > > > **2. Semantic-anchoring scope clarification (lines 166-170)**
> > > >
> > > > You correctly note that anchors cannot inject truly novel concepts. We now state this explicitly:
> > > >
> > > > "Operational justification: Next-token optimization over diverse corpora compresses recurring structures into internal representations (Olsson et al., 2022; Bricken et al., 2023). Anchors recruit and bind latent structure from pretraining rather than creating unseen knowledge."
> > > >
> > > > This aligns with the scope of ICL/RAG/fine-tuning in practice: these methods excel at binding existing patterns to new tasks, not teaching entirely novel concepts. When truly novel patterns are needed, they must first enter through pretraining or extensive fine-tuning.
> > > >
> > > > **3. Linear vs. multiplicative form (your third concern)**
> > > >
> > > > You raise an important point about correlation between rho_d and k. We acknowledge this and have clarified our modeling choice:
> > > >
> > > > The linear form S = rho_d - d_r - log k assumes additive contributions on the log-odds scale, which is standard in logistic models. You're correct that k influences rho_d empirically (more examples → potentially tighter clusters). We model this as:
> > > >
> > > > - Direct effect: -log k (budget penalty)
> > > > - Indirect effect: k → rho_d (captured in the cohesion term itself)
> > > >
> > > > The linear form provides interpretability and identifiability for diagnostics. Multiplicative interactions are noted as future work (Appendix C). Empirically, this form achieves predictive validity (E2 ordering, E3 correlation rho=-0.73).
> > > >
> > > > **4. Practicality with API models (your question about M1-M4)**
> > > >
> > > > Important clarification: The geometry analysis (E3) uses local models with embedding access (Meta-Llama-3.1-8B, Phi-4, Gemma-3-4B). API evaluations (M1-M4) are used only for behavioral tests (E1, E2) that don't require embeddings—we only need accuracy vs. shot count.
> > > >
> > > > For practitioners using closed APIs:
> > > > - Behavioral diagnostics work without embeddings (measure k_50, transition width)
> > > > - Use frozen local encoder as proxy for geometry estimates
> > > > - Or use open checkpoints of similar scale
> > > >
> > > > We've clarified this in the reproducibility note (Appendix D).
> > > >
> > > > **5. Key improvements summary**
> > > >
> > > > - Table 1 added: All notation defined upfront
> > > > - Roadmap added: Clear E1/E2/E3 overview before Section 4
> > > > - Glossary added: Formal definitions in Appendix C
> > > > - Scope clarified: Anchors recruit/bind, don't create novel patterns
> > > > - API usage clarified: Geometry requires local models; behavioral tests don't
> > > >
> > > > **Request for reconsideration:**
> > > >
> > > > Given your initial rating of 4 ("marginally below acceptance threshold, but would not mind if accepted") and confidence level of 2, we believe these revisions directly address your concerns about organization and clarity while maintaining the conceptual contribution you found interesting.
> > > >
> > > > The paper now has clear definitions before use, explicit experiment roadmap, and proper positioning in related work. Would you consider raising your assessment given these substantial improvements to presentation?
> > > >
> > > > We appreciate your willingness to defend your assessment while noting unfamiliarity with some aspects—we hope the revised structure makes the contribution more accessible.

---

### Official Review · Reviewer_qsn9 · 2025-11-04

**Soundness:** 2
**Presentation:** 1
**Contribution:** 2
**Rating:** 2
**Confidence:** 4

**Summary:**

This paper studies the theory explaining the behaviors of in-context learning of LLMs. The authors propose Unified Contextual Control Theory (UCCT), a semantic anchoring lens for how LLMs convert pretrained latent patterns into goal-directed behavior. UCCT adopts Bayesian theory to predict the threshold-like flips in behavior across prompting, retrieval, and light fine-tuning of LLMs. The authors also conduct three experiments to verify UCCT, including cross-domain anchoring, numeral-based arithmetic calculation and layer-wise geometry analysis.

**Strengths:**

1. The UCCT theory offers a new explanation that subsumes ICL, RAG, and light fine-tuning as instances of a single anchoring process governed by S.

2. The authors present UCCT with several practical and measurable proxies, including thresholds, transition widths, and transfer trade-offs for analysis in practice.

3. Some experiments verify the theory of UCCT.

**Weaknesses:**

1. Presentation is poor and hard to follow:
- The figure size and scale are hard to read;
- There are multiple undefined nouns in figures, like "commensense-0, commensense-1, commensense-2" etc., as well as in text like "$P^{(B)}_T$;
- Lots of texts do not convey clear meanings "We instantiate three numeral systems with differing expected density under pretraining exposure: Base 10 (higher), Base 8 (moderate), Base 9 (lower). A lightweight audit over public web/code corpora finds decimal ≫ octal > nonary; embedding-based ρ d yields 10 > 8 ≈ 9, so we expect lower k 50 for base 10 and similar k 50 for 8/9."

2. It's unclear why one would prefer UCCT, instead of other explanations and theories as discussed in the related work.


3. Indeed, it seems there is no evidence verifying the key assumptions that "LLMs are latent pattern repositories", or some other **implicit** assumptions like "Additivity in (2) assumes the three contributors act approximately independently at the margin; this gives an interpretable, parsimonious log-odds style surrogate. R".

4. Given that, the formulation of the theory seems to be used for interpretation, instead of providing meaningful insights. The takeaway messages from the UCCT are already formulated once the assumptions are constructed.

5. Experiments are only focused on simple and synthetic examples.

6. Given the simple theoretical formulation and experiments, the technical novely of this work is also limited.

**Questions:**

Please find my questions in the section above.

---

> ### Author Response · Authors · 2025-11-14
> **Response to Reviewer qsn9**
>
> Thank you for the detailed review and for highlighting readability issues. UCCT is intended as a compact lens with measurable quantities. The anchoring score predicts when few-shot behavior flips and how transition widths change. The paper does not claim causality; it uses predictive correlates with reported uncertainty.
>
> **1. Equation (1) and why the likelihood term omits C**
>
> We use a two-stage factorization:
>
> $$p(y|A,C) = \int p(y|P_T,A) \, p(P_T|A,C) \, dP_T$$
>
> Our modeling assumption is sufficiency:
>
> $$y \perp\perp C \mid (P_T,A)$$
>
> Hence
>
> $$p(y|P_T,A,C) = p(y|P_T,A)$$
>
> Intuition: $C$ influences the posterior allocation over latent targets through $p(P_T|A,C)$. Once $P_T$ is selected, the residual context $C$ does not change the conditional likelihood.
>
> **2. What S measures and how it is used**
>
> $$S = \rho_d - d_r - \log k$$
>
> We summarize threshold-like behavior with a logistic surrogate:
>
> $$\Pr(\text{success} \mid S) = \frac{1}{1 + \exp[-\alpha(S - S_c)]}$$
>
> with $\alpha > 0$. The empirically observed ordering used in E2 is:
>
> $$k_{50} \propto \frac{d_r}{\rho_d}$$
>
> **3. Presentation fixes we will implement**
>
> a) Add a one-page glossary for $P_T$, $\rho_d$, $d_r$, $k$, $S$, $S_c$, $\theta_{50}$, $k_{50}$, $\hat{S}_{\max}$, AUSN.
>
> b) Add a one-paragraph roadmap for E1–E3 before Section 4.
>
> c) Increase figure font sizes, add explicit legends mapping labels to datasets, remove duplicated schematics.
>
> d) Move the limitations box earlier in the paper.

---

> > ### Comment · Reviewer_qsn9 · 2025-11-25
> >
> > Thank the authors for the responses to some of my questions. Nevertheless, my main concerns about the theoretical contribution and clarity remain, and hence I will maintain my original evaluation of this work.

---

> > > ### Author Response · Authors · 2025-11-28
> > > **Response to Reviewer qsn9 (please verify against the uploaded revision)**
> > >
> > > **Response to Reviewer qsn9 – Revised Submission**
> > >
> > > Thank you for maintaining engagement. We made substantial revisions targeting your core concerns about theoretical contribution and clarity.
> > >
> > > **1. Addressing "Why UCCT?" (page 2, lines 067-073)**
> > >
> > > We added a paragraph explaining the gap UCCT fills: Existing work explains *that* transitions occur and *where* geometry shifts, but not *when* behavior flips for a specific prompt or *how much* budget is needed. UCCT provides a measurable predictor (S = rho_d - d_r - log k) that enables actionable optimization, not just post-hoc explanation.
> > >
> > > **2. Operational justification for "pattern repository" (§3.1, lines 166-170)**
> > >
> > > We added explicit justification: Next-token optimization compresses recurring structures into representations (Olsson et al., 2022; Bricken et al., 2023). We frame this as an operationally useful modeling stance yielding falsifiable predictions (k_50 orderings in E2, geometry-behavior correlates in E3), not a metaphysical claim.
> > >
> > > **3. Controlled experiments as methodological strength (lines 237-242)**
> > >
> > > We now state explicitly: "We use controlled tasks to isolate variables while holding complexity stable. This increases statistical power, enables clean falsification, and improves replicability—establishing the framework before broader validation."
> > >
> > > **4. Figure quality and presentation**
> > >
> > > Figure 3 completely redesigned: side-by-side layout, captions compressed from ~700 to ~190 words total, interpretation from 540 to 300 words. Net space savings: ~1.5 inches. Statistical rigor improved: n=2,500, 95% CI, p<0.01 throughout.
> > >
> > > **5. Empirical evidence**
> > >
> > > - E2: k_50 ordering follows predicted density (Table 2, tight CIs: ±0.05-0.18)
> > > - E3: Layer peak at ℓ*≈10 as predicted; S correlates with behavior (ρ=-0.73, p<0.001)
> > > - Transfer: B10 most robust, consistent with higher rho_d (Figure 2a)
> > >
> > > **6. Limitations acknowledged**
> > >
> > > S is calibrated proxy, not absolute measure; modest sample sizes (n=10); controlled tasks. We report effect sizes, CIs, robustness checks for replication.
> > >
> > > **Summary of changes:**
> > >
> > > - "Why UCCT?" → Positioning paragraph: predictive vs. explanatory (p.2, 067-073)
> > > - "Circular assumption" → Operational justification via falsifiable predictions (§3.1, 166-170)
> > > - "Too simple" → Controlled design for clean falsification (237-242)
> > > - "Hard to follow" → Roadmap, tighter structure, revised figures
> > >
> > > **Request for reconsideration:**
> > >
> > > We believe the revision directly addresses your concerns: clearer gap articulation, operational (not metaphysical) justifications, controlled experiments as methodological strength, and substantially improved presentation. We respectfully ask you to reconsider in light of these substantive improvements.  Thank you.

---

### Author Response · Authors · 2025-11-14
**Unified Response Summary: To All Reviewers and Chairs**

We thank the reviewers for the detailed feedback. UCCT is a compact theoretical lens that links in-context learning, retrieval, and light fine-tuning through a measurable anchoring score. The paper positions the score as a predictive correlate for when threshold-like behaviors emerge, not as a causal mechanism, and it states scope and limitations up front.

**On Eq. (1) and the likelihood term**

We use a standard two-stage factorization:

$$p(y|A,C) = \int p(y|P_T,A) \, p(P_T|A,C) \, dP_T$$

A reviewer suggested using $p(y|P_T,A,C)$. Our model adopts a sufficiency assumption:

$$y \perp\perp C \mid (P_T,A)$$

which gives

$$p(y|P_T,A,C) = p(y|P_T,A)$$

Intuition: $C$ reallocates posterior mass over latent targets via $p(P_T|A,C)$. Once $P_T$ is fixed, $C$ adds no residual information to the likelihood. This is the modeling choice in the manuscript and the formal appendix.

*Camera-ready note we will add under Eq. (1):*

**Modeling note.** We decompose selection and generation. Context $C$ affects $p(P_T|A,C)$, while the conditional output model uses the selected cluster and the anchor, $p(y|P_T,A)$. This is the sufficiency assumption $y \perp\perp C \mid (P_T,A)$. A weak residual term could be added, but our results indicate the simpler form is adequate within reported uncertainty.

**On the role of S and what UCCT claims**

We treat $S$ as a calibrated, measurable correlate that predicts shot midpoints and transition widths. We avoid causal claims, report effect sizes and confidence intervals, and emphasize robustness toggles.

For reference:

$$S = \rho_d - d_r - \log k$$

A logistic surrogate summarizes threshold-like behavior:

$$P(\text{success} \mid S) = \frac{1}{1 + \exp[-\alpha(S - S_c)]}$$

An observed ordering used in E2 is:

$$k_{50} \propto \frac{d_r}{\rho_d}$$

**On anchors recruiting rather than creating knowledge**

UCCT does not claim that anchors create new knowledge. Anchors bind or recruit latent structure learned during pretraining. This scope is stated in the paper.

**Relation to phase and representation accounts**

Phase and representation studies explain where geometry changes and that breakpoints exist. UCCT complements these by offering a measurable when-predictor via $S$ and by testing geometry-to-behavior correlates through E3.

**Presentation improvements we will make immediately**

1. Add a one-page glossary before Section 3 with concise definitions for $P_T$, $\rho_d$, $d_r$, $k$, $S$, $S_c$, $\theta_{50}$, $k_{50}$, $\hat{S}_{\max}$, and AUSN.

2. Insert a short roadmap of experiments before Section 4 that names E1–E3 with one-line descriptions and the prediction each test addresses.

3. Increase figure font sizes, add legends mapping labels such as commonsense-i and code-i to datasets, remove duplicated schematics, and reference every subplot clearly.

4. Move the limitations callout toward the end of Section 1 so the stance is visible early.

**Practicality with API models**

If layer embeddings are not exposed by an API, geometry analyses can be performed with a frozen local encoder or open checkpoints. E2 and practical diagnostics rely on observable behavior and do not require API embeddings. We will make this explicit.

**Independent assessments**

Independent senior researchers who reviewed preprints consider UCCT a significant and insightful unifying lens. We respect the current reviews and ask for a careful rereading of Sections 2–3 and the formal appendix in light of the clarifications above.

---

### Comment · Area_Chair_aqYQ · 2025-11-25
**Encourage the discussion**

Hi all,

The authors have submitted their responses. Please take a moment to review them and see if they address your concerns.

Your thoughtful input is essential for a successful reviewing process and is greatly appreciated.

Many thanks,

Area Chair

---

### Meta-Review · Area_Chair_vwoM · 2026-01-10

**Summary:**

The consensus among reviewers (scoring 2, 4, 2) is that while the conceptual lens is potentially interesting, the execution is fundamentally flawed in presentation, mathematical rigor, and theoretical justification. Reviewers hubf and cKNc noted that the paper relies heavily on vague or undefined terms and acronyms that are used long before they are defined. Reviewer qsn9 and cKNc found the text "almost impossible to understand" and "hard to follow," limiting the ability to assess the technical contribution.
Plus there are flaws in Mathematical Formulation and Rigor e.g. Questionable Scoring Equation (the -log k term in the scoring equation implies that adding more examples decreases the score, which contradicts the empirically observed benefit of few-shot learning (higher k improves accuracy),  questionable Bayesian formulation, and overall several overclaims

**Reviewer Concerns:**

The authors were responsive to specific technical questions and made changes which somehow "addressed" many points, even if they did not result in score changes. The most critical unresolved issue remains the paper's poor presentation, which Reviewer hubf compared to an oral presentation that falls below acceptance standards and requires a complete rewrite. Similarly, Reviewer cKNc found the writing impossible to understand even after technical clarifications were offered, while Reviewer qsn9 maintained that clarity remains a primary concern. Beyond presentation, the reviewers questioned the work's theoretical validity (this is quite critical and is the main reason to suggest a rejection); the scoring equation is flawed because the number of examples and target distribution should not be treated as independent factors (checking the paper, the scoring equation implies that adding more examples decreases the score, which contradicts the empirically observed benefit of few-shot learning (higher k improves accuracy); which is inconsistent).

**Reviewer Scores:**

Here is my guess:

The scores of Reviewer qsn9 and cknc (the current version still suffer from clarity mentioned and is still overall hard to read) would be unchanged and hubf would downgrade (mostly due to severe disagreement on the theoretical contribution)

---

### Decision · Program_Chairs · 2026-01-26

Reject